# Emergence of task-related spatiotemporal population dynamics in transplanted neurons

Harman Ghuman[1,3], Kyungsoo Kim[1,3], Sapeeda Barati[1] & Karunesh Ganguly [1,2] ✉

Loss of nervous system tissue after severe brain injury is a main determinant of poor functional recovery. Cell transplantation is a promising method to restore lost tissue and function, yet it remains unclear if transplanted neurons can demonstrate the population level dynamics important for movement control. Here we present a comprehensive approach for long-term single neuron monitoring and manipulation of transplanted embryonic cortical neurons after cortical injury in adult male mice performing a prehension task. The observed patterns of population activity in the transplanted network strongly resembled that of healthy networks. Specifically, the task-related spatiotemporal activity patterns of transplanted neurons could be represented by latent factors that evolve within a low dimensional manifold. We also demonstrate reliable modulation of the transplanted networks using minimally invasive epidural stimulation. Our approach may allow greater insight into how restoration of cell-type specific network dynamics in vivo can restore motor function.

Neurological impairments caused by acquired brain damage due to stroke or traumatic brain injury are directly the result of lost neurons and connections. Such loss of nervous system tissue can impair the ability of motor cortical regions to generate the complex spatiotemporal dynamics that drive movement control[1–8]. For example, primary motor cortex (M1) activity demonstrates highly stable sequential activity patterns that appear to propagate to and drive downstream structures[3,4,9–11]. Such spatiotemporal activity patterns can also be represented by a few "latent factors" that evolve within a low dimensional space ('manifold')[4,6,7]. Recent work further indicates that recovery of neural sequences and latent factor activity is correlated with recovery of motor function[1,2,12]. Notably, modulation of task-related neural co-firing, which appears to augment latent factor activations, can also improve function if there is residual tissue and partially reconstituted activity patterns[2,12].

Unfortunately, in a significant subset of cases there is simply not sufficient nervous system tissue for either rehabilitation or neuromodulation to support recovery[13]. Unlike in rodents, the endogenous capacity of the human brain to regenerate after injury is limited[14–18].

Cell transplantation is a promising method and has been extensively evaluated to compensate for lost tissue[19–24]. However, a major challenge for the field is to develop targeted methods for in vivo long-term monitoring and modulation of transplanted neurons as they integrate into damaged host networks. Past methods have largely relied on ex vivo electrophysiological and histological methods;[25–27] these methods do not provide sufficient spatiotemporal resolution to track activity patterns with recovery. While in vivo electrophysiology methods has also been used[28,29], they cannot specifically identify host versus donor cells. This is because large numbers of host newborn neurons are known to migrate to the site of injury in rodents[30,31], including to the site of cell implantation[32–34]. Thus, current methods do not allow specific long-term tracking of transplanted neurons.

Here we present a comprehensive toolbox for long-term monitoring and manipulation of transplanted neurons after motor cortical injury in freely behaving adult mice. We used an implantable miniscope system for long-term calcium imaging to monitor transplanted embryonic neurons during a skilled prehension task. We specifically tracked transplanted neurons which expressed genetically encoded

[1]Department of Neurology, University of California, San Francisco, San Francisco, CA, USA. [2]Neurology Service, San Francisco Veterans Affairs Medical Center, San Francisco, CA, USA. [3]These authors contributed equally: Harman Ghuman, Kyungsoo Kim. ✉e-mail: karunesh.ganguly@ucsf.edu

calcium indicators, thus easily distinguishing them from the host tissue. This allowed us to test whether transplanted neurons after a motor cortex stroke can demonstrate the population dynamics associated with movement control. While transplanted neurons and organoids can survive long periods and even demonstrate evoked sensory responses[20,35–39], they have been reported to demonstrate activity patterns more consistent with immature networks (i.e., highly synchronized bursting activity) for prolonged periods[38,40,41]. Similar burst activity patterns are known to persist in isolated networks such as cultured neurons[42,43]. Thus, it remains unclear if transplanted neurons can demonstrate the sequential activation and co-firing patterns associated with movement control.

Remarkably, we could reliably track the emerging dynamics of transplanted neurons from an initial uncoordinated state to a highly reliable state of spatiotemporal activation that was temporally locked to grasping movements. The long-term evolution of activity patterns was evident during task-specific rehabilitation training and was correlated with improvements in performance; the resulting patterns of population activity closely resembled that of healthy motor cortical circuitry. Notably, our toolbox could also be used to monitor intra-graft blood flow dynamics and demonstrate how physical proximity of blood vessels predicted cell survival. Lastly, we demonstrate how our platform allows reliable modulation of transplanted neural networks using minimally invasive epidural stimulation, a feature that will eventually allow closed-loop stimulation in an activity-dependent manner. Together, our approach will allow for greater insight into how neural transplants integrate into injured networks and restore network dynamics in a targeted manner.

## Results

We tracked the calcium dynamics of transplanted embryonic cortical neurons from donor mice expressing GCaMP6f—under either the Syn1 or CamKIIα promotor (Fig. 1a)—during circuit maturation and integration in an adult injured host brain covering primary motor cortex (M1) using a head-mounted miniscope (Fig. 1b). Placement of the 1 mm x 1 mm PRISM lens allowed us to record the activity of transplanted neurons across the cortical mantle (Supplementary video 1). One week before surgery, mice were habituated to an automated behavioral box[44], and then food-restricted to determine their preferred handiness using the reach-to-grasp motor task. Since transplanted neural progenitors and neurons have been shown to regulate motor function post-stroke[20,22,24], our efforts here primarily focused on the emergence of task-related neural dynamics in transplanted neurons. After two weeks, transplanted mice performed daily motor training (Fig. 1c), leading to significant improvements in behavioral performance over the weeks of monitoring (Fig. S1a). Calcium imaging during the reaching task began when transplanted neurons were initially detected using the miniscope, around 2–4 weeks post-implantation (WPI). We continued monitoring activity for a total of 12 weeks. We then performed epidural stimulation and blood-flow experiments to manipulate the transplanted network dynamics and monitor intra-graft blood flow dynamics.

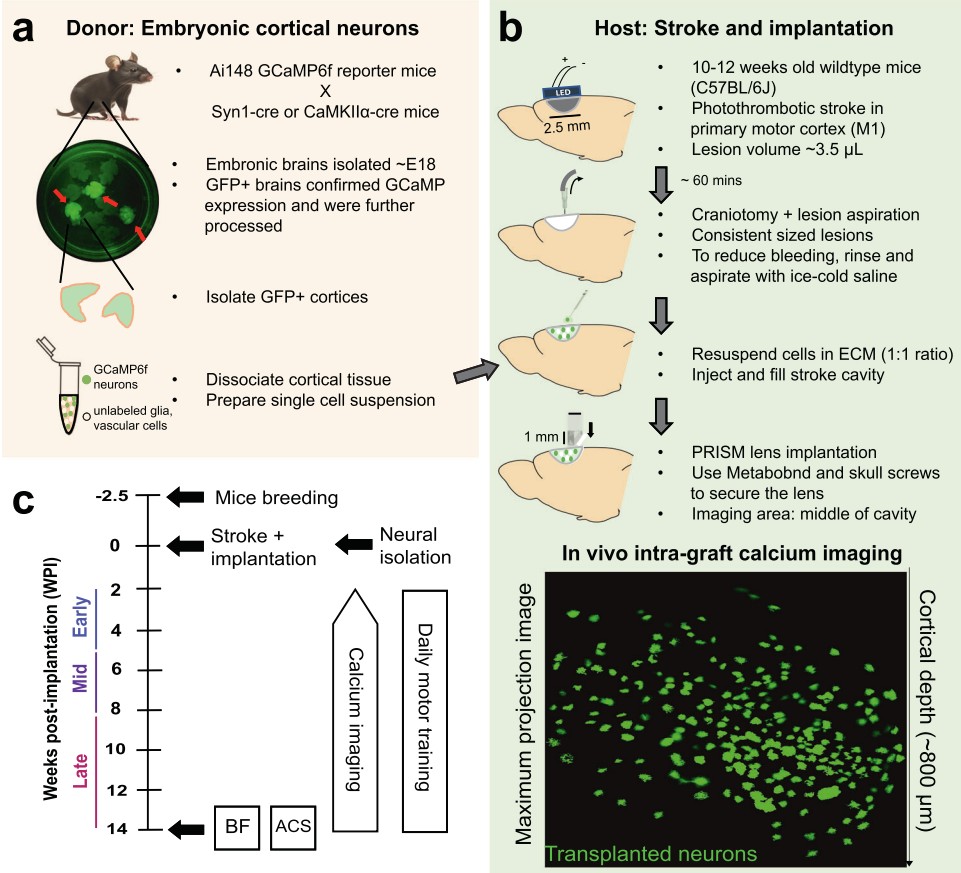

**Fig. 1 | Experimental setup and timeline. a** For donor neurons, we utilized cre-dependent GCaMP6f expression in embryonic cortical neurons by crossing transgenic mice with pan-neuronal (i.e., Syn1 or CaMKIIα) cre expression with GCaMP6f reporter mice (Ai148). GFP+ cortices were dissociated to obtain a single cell suspension of embryonic cortical cells including GCaMP6f neurons, glia and vascular endothelial cells. **b** For host mice, we used 10–12 weeks old stroke wildtype C57BL/6J mice. Prior to cell injection, host mice received unilateral stroke in primary motor cortex (M1) contralateral to dominant forepaw. Isolated cells were resuspended in ECM hydrogel and injected into the cavity. A1 x 1 mm PRISM lens was then inserted. **c** Timeline of the experiments.

## Movement-related neural activation and network modulation

We first tested the hypothesis that implanted embryonic neurons can develop motor task-specific activity with long-term motor training (Fig. 2a). To then investigate whether transplanted neurons generated reliable responses during forelimb movements, we acquired calcium images while implanted mice performed the prehension task. We then aligned all the neural data to the moment of pellet touch (Supplementary video 2). Identification of task-modulated neurons was determined using an ANOVA test (i.e., is the pattern of activity significantly modulated during task periods, see methods). This allowed us to categorize neurons as task modulated or task non-modulated (Fig. 2b). We then found evidence for greater proportions of task modulated neurons over time (Fig. 2c). Given that there is sequential firing of intact M1 neurons during motor tasks[1,2,10] (Fig. 2d), we also analyzed the temporal activation of modulated neurons over time. The transplanted neurons demonstrated robust single trial and trial averaged sequential activation and co-firing associated with pellet touch.

We found that over the course of motor training, the total number of task-modulated neurons (Fig. S1b), as well as the proportions of task-modulated neurons in the network increased (Fig. 2e, one-way ANOVA: $p = 0.024$, $R^2 = 0.638$; t-test: $p = 0.0042$), with a peak modulation of ~49% in the late period, which was not statistically different than neurotypical mice. Notably, we measured task-modulated neurons across both the promoter groups (i.e., Syn1 vs. CamKIIα) and found no significant differences (unpaired t-test: (early) $p = 0.73$, (mid) $p = 0.41$, (late) $p = 0.67$). This increase in cortical depth-independent (Fig. S1c), task-modulated population may reflect the greater synaptic coupling among transplanted and host neurons[35,39,45]. Interestingly, a subset of transplanted neurons (~6% of total neurons) showed preferential activation during the door open cue tone before reach onset, suggesting that these neurons also received inputs related to the sensory cues that inform movement onset (Fig. S1d).

To better understand the relationship between task-related activity and performance, we calculated the correlation between success rate and the proportions of task-modulated neurons during task training after stroke. We found that performance was correlated with the proportions of task-modulated neurons in each of the transplanted networks (Fig. 2f, linear mixed-effect model (LME): $p$: 0.002, r: 0.6). Interestingly, the extent of task modulation in the transplanted neurons was also found to be correlated to task attempt rate (Fig. S1e), emphasizing that repeated skilled motor training enhances task-related activity in the graft. Together, these findings indicate that in mice performing daily task-training, transplanted neurons can indeed produce reliable task-related activity that is significantly modulated during movement preparation and execution, and the proportions of movement responsive neurons reflects the behavioral performance after stroke.

## Emergence of movement-related population dynamics

To further understand the population level dynamics underlying transplanted neural activity during task performance, we performed factor analysis (FA) and measured the shared-over-total variance (SOT)[12,46]. SOT is the ratio of the neural variance associated with population coupling ('shared variance') to the neural variance associated with decoupled firing, i.e., isolated activation of a neuron or 'private variance'. The SOT should increase if there is greater population task-related co-firing. We thus transformed population activity into private and shared signals to disentangle how independent and coordinating sources of neural variance changed over the course of network integration (Fig. 3a). We first used principal component analysis (PCA) to determine the variance accounted for; we found that the first two PC dimensions explained ~90% of the neural variance (Fig. S2). The PCA result implied that two linearly independent shared factors can explain most of neural activity. We thus measured SOT using two factors. FA with two shared factors revealed that the SOT gradually

increased during the task period over time (Fig. 3b, one-way ANOVA: $p = 0.016$, $R^2 = 0.692$; early = $45.0 \pm 3.1$SEM, mid = $54.8 \pm 2.4$SEM, late = $67.5 \pm 3.6$SEM, healthy control = $64.1 \pm 5.6$SEM). Prominently, the shared variance of the first two factors from the transplanted network activity was not statistically different from that of healthy cortical circuitry. This indicates that transplanted neurons significantly increased their task-related co-firing during middle to late sessions and displayed similar co-firing levels as neurotypical mice.

The similarity in task-related dynamics in late transplanted and intact cortical networks was then visualized via Gaussian-process smoothening, which projects the neural responses to a low-dimensional space where task-related 'neural trajectory' can be traced through time. While early trial-averaged neural trajectories showed poor rotational dynamics during the task, late transplanted networks followed trajectories commonly seen in healthy intact cortical networks (Fig. 3c). Similar trends were observed across all transplanted mice (Fig. S2e). This suggests that transplanted neurons can indeed display similar population level co-firing and rotational dynamics as intact networks.

## Tracked single neurons show increasingly correlated movement related activity

One of the key advantages of in vivo imaging is the ability to observe and record the same brain region for extended periods and to track long-term changes. To understand the integration process, it is necessary to track the same cells throughout the integration process in the living brain. This methodology can substantially reduce the number of experimental animals needed. However, there is no prior report of in vivo tracking of transplanted neurons at a single cell resolution in a behaving animal. We leveraged our unique ability to track individual transplanted neurons longitudinally over sessions spanning 3 months (Fig. 4a). Although we were unable to track all of the transplanted neurons, likely because of tissue remodeling following stroke, we reliably tracked around 15% of total neurons for 3 months (Fig. S3a). The number of tracked neurons across 2 months (Early-Mid or Mid-Late) was significantly higher at ~30% of total neuronal population.

To understand how single neurons changed their activity patterns over time, we looked at the firing patterns of tracked neurons during and outside the task window. This revealed that during the early phase, most neurons co-fired but were not robustly behaviorally related (Fig. 4b). During middle to late phases, neurons became selectively more active during the task. Trial-averaged task activity from an example tracked neuron shows this neuron became movement responsive during the late phase (Fig. 4c). Trial to trial activity from the tracked neurons reveals that with time, transplanted neurons increasingly became active during the forelimb movement (Fig. 4d). To quantify this increase in task co-activation, we performed cross-correlations of tracked neural activity. Each circle corresponds to a neuron and a line connecting neurons indicates significant covariance between 2 neurons during the motor task (Fig. 4e). Our analysis revealed that the transplanted network strongly co-varied during the late period. While many neurons displayed an increase in firing during forelimb movements, others showed a significant reduction of activity during the task, resulting in negative correlations (Fig. 4f). Notably, the average calcium peak fluorescence from active neurons did not significantly change over time (Fig. S3b, one-way ANOVA: $p = 0.11$, $R^2 = 0.47$; paired t-test: n.s. for all pairs). Similar trend was evident in tracked neurons from healthy trained mice (Fig. S3c, d). Interestingly, such sparsification of task related activity can be observed in intact motor networks during long-term learning[47].

## Modulation of neural activity in transplanted networks

Studies in awake and anesthetized animals have found that externally applied electrical fields can boost firing rates and bias spike timing[12,48], but it is unclear how externally applied electric fields interact with

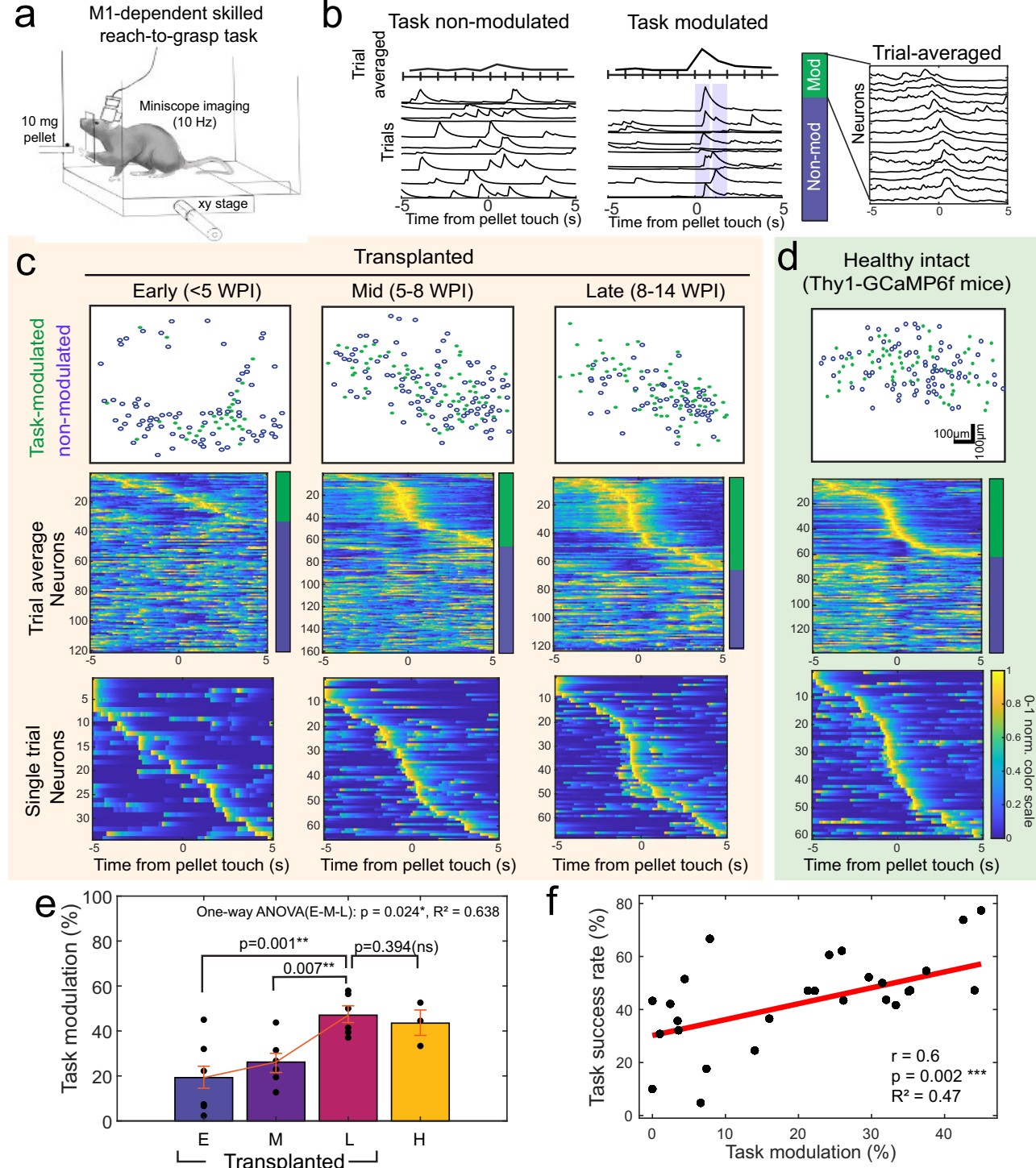

**Fig. 2 | Movement-related neural activation and network modulation. a** Images of transplanted neurons were collected while mice performed the reach-to-grasp prehension task. **b** Neurons were categorized as task-modulated and non-modulated based on trial-to-trial consistency (i.e., significant ANOVA test). **c** Transplanted neurons demonstrated robust single trial and trial-averaged sequential activation and co-firing during the task and the proportions of task-modulated neurons increased over time. **d** Proportions of task-modulated neurons during task in intact healthy Thy1-GCaMP6f mice. **e** Proportions of task-modulated transplanted neurons (E: 19.6%, M: 26.6%, L: 48.9%) in comparison to

neurotypical mice (one-way ANOVA: $p = 0.024$, $R^2 = 0.638$; one-tailed paired t-test: ns $p > 0.05$, and **$p \leq 0.01$, $n = 6$ transplanted mice and $n = 3$ neurotypical mice). All data are presented as the mean ± standard error of the mean (SEM). Source data are provided as a Source Data file. **f** Correlation between behavioral performance and the proportions of task-modulated neurons in transplanted networks (linear mixed-effect model (LME), $p$: 0.002, r: 0.6, $R^2$: 0.47). Each dot represents a single behavioral session. All neural activation heat maps share the same 0-1 normalized color scale (see details in methods). Source data are provided as a Source Data file.

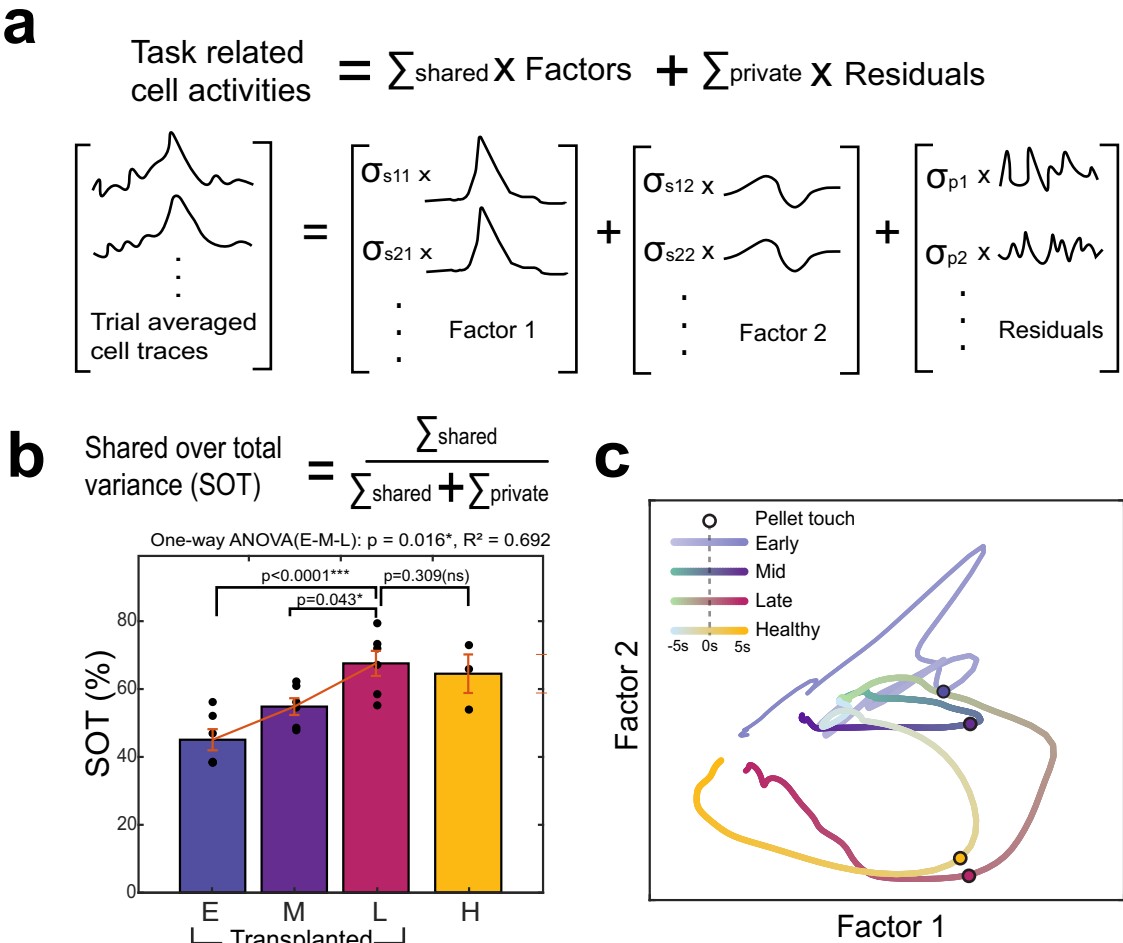

**Fig. 3 | Emergence of movement-related population dynamics. a** We performed factor analysis (FA) and transformed population activity of the transplanted neurons into private and shared signals to measure shared-over-total variance (SOT). **b** Our analysis revealed that the SOT gradually increased during the task period over time (one-way ANOVA: $p = 0.016$, $R^2 = 0.692$; t-test: ns $p > 0.05$, *$p \leq 0.05$, and ***$p \leq 0.001$ one tailed paired t-test in E-M-L and one tailed unpaired t-test between L and H, $n = 6$ transplanted mice, $n = 3$ healthy mice) and the shared variance

explained by first two factors from the late transplanted network activity was similar to that of healthy cortical circuitry. All data are presented as the mean ± SEM. Source data are provided as a Source Data file. **c** Trial-averaged projections of transplanted and healthy network activity during the task window revealed similar rotational dynamics of transplanted neurons. Circle indicates moment of pellet touch.

---

transplanted networks and influence their internal dynamics. Such stimulation can also be used to monitor transplanted neuronal viability and functional properties. For example, are populations of transplanted neurons that were not movement responsive reliably modulated with stimulation? To answer these questions, we applied low frequency alternating current stimulation (ACS) covering both the transplanted network as well as surrounding host tissue (Fig. 5a). Calcium activity from the transplanted neurons was recorded for a total of 15 min, including baseline, ACS and post-ACS periods, and time-locked to ACS onset.

Trial average neural activity revealed that around 75% of the transplanted neurons showed significant responses to ACS (Fig. 5b). We found that ACS acutely enhanced the neural dynamics of the transplanted network to increase network co-firing or synchrony (Fig. 5c, Supplementary video 3). This acute entrainment of neural activity led to a significant increase in the number of detected single events and co-firing during the entire 5 min stimulation period when compared to baseline activity (Fig. 5d, paired t-test: Baseline-ACS $p = 0.0063$, ACS-post and Baseline-Post $p < 0.0001$). This increase in excitability in the transplanted network during ACS was then followed by a period of reduced neural activity, with ~70% of the neurons showing reduced activity post-ACS compared to baseline (Fig. 5e).

Similar long-lasting inhibitory responses are commonly seen after intracortical stimulation in healthy cortical circuits[49-51], and suggests that the transplanted network is mature and capable of stabilizing network excitability. Analysis of cortical-depth dependent modulation reveals ACS-modulated neurons across the cortical mantle (Fig. S4). This suggests that although the currents are applied on the cortical surface, the generated electrical fields can reliably modulate neurons across all cortical layers.

To further probe functional interactions of the transplanted neurons, we matched single neurons from ACS and behavioral sessions (Fig. 5f). This allowed us to group the population of transplanted neurons based on their interactions with the perilesional cortex: behavior and ACS modulated (Beh+, ACS+), behavior modulated only (Beh+, ACS-), ACS modulated only (Beh-, ACS+), and non-responsive neurons (Beh-, ACS-). Interestingly, two-thirds of the neurons that were not task-modulated were still responsive to ACS. The transplanted network also displayed equal proportions (~25%) of non-responsive neurons and (Beh+, ACS+) modulated neurons. In summary, stimulation allowed us to reliably manipulate the transplanted networks and assess overall neural viability relative to task responsiveness. Importantly, this approach can be used to further perform activity-dependent or closed-loop stimulation to modify network integration.

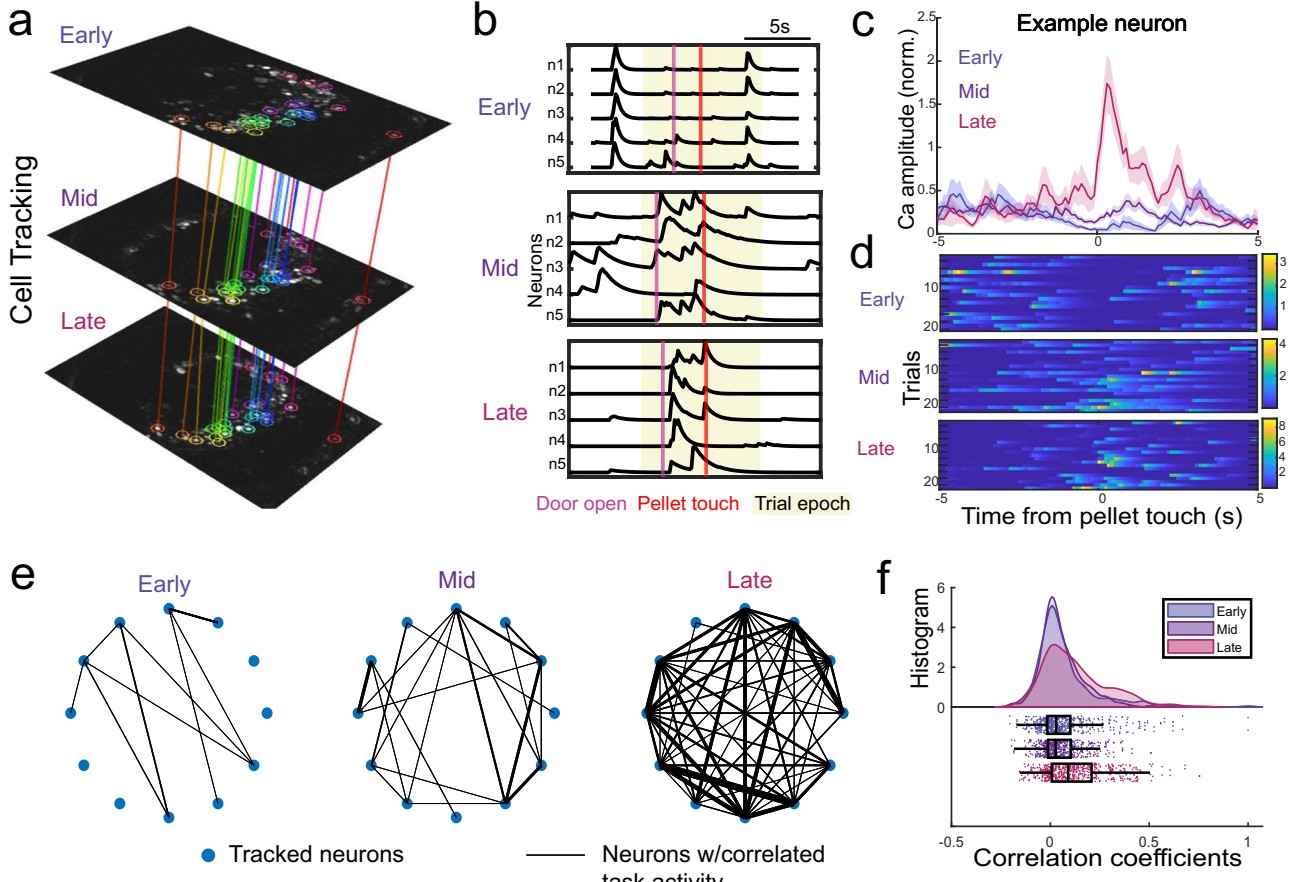

**Fig. 4 | Long-term single cell tracking of transplanted neurons. a** Using the CalmAn cell registration tools (and manual inspection), we tracked a subset (~15% of total) of individual neurons longitudinally across 3 months. **b** During the early phase, most neurons co-fired outside of the task window. The same neurons became selectively more active during the task over time. **c** Trial-averaged task activity (trial average activity ± SEM), and **d** trial-to-trial activity of a single transplanted neuron, it became active during forelimb movements. **e** Illustration of task related cross-correlations. We connected tracked neurons (blue circles) with lines that showed significant task covariance. **f** Transplanted networks strongly co-varied during the late period. In the late period, there were both positive and negative correlations (i.e., reductions in co-firing). The histogram box plot displays median and percentile boundary (i.e., 25th and 75th at the box edges) and whiskers are 1.5 interquartile range. $n = 570$ tracked cell correlation from 6 transplanted mice. Source data are provided as a Source Data file.

## In vivo microcirculatory dynamics within the transplanted graft

Direct injection of cells into the stroke cavity, which does not have any blood vessels, exposes cells to prolonged periods of hypoxia[52,53], and has been understood to be one of the major reasons of poor cell survival and engraftment post-implantation[54–56]. Thus, real time measurements of blood flow changes inside the implanted graft can aid in identifying functional blood vessels and microcirculatory dynamics underlying successful cell survival and integration.

To determine whether the graft vessels perfused blood, we performed an intraperitoneal injection of fluorescein-dextran at 14 WPI and imaged blood flow in real time within the graft (Fig. 6a, Supplementary video 4). Notably, this was done during the late period; attempts to do this in the early period was not successful due to extensive leakage of dye, likely because of incompletely formed vessels. In the late period, all implanted grafts that were imaged had vessels and erythrocytes circulating through them. To better understand the relationship between surviving neurons and blood flow, we co-registered the calcium imaging and blood flow videos (Fig. 6b). Based on our understanding that most cells are within 200 μm from the nearest vessel (oxygen diffusion limit)[57,58], we hypothesized that surviving transplanted neurons must be within this range. Analysis of the distance between transplanted neurons and the nearest blood vessel revealed that 95% of the surviving neurons from all grafts resided within 76.2 μm of a functional blood vessel (median = 13.46 μm, 25th percentile = 3.7 μm, and 75th percentile = 31.83 μm).

Based on temporal cross-correlation analysis, we determined the blood flow speed in vessels within the implanted graft (Fig. 6c). This revealed an average blood flow speed of ~1 mm/s, which is similar to the reported blood flow speed in healthy cortical tissue[59]. These results highlight that the vessels forming within the graft have circulating erythrocytes and thus are likely transporting oxygen and nutrients to the transplanted network. The ability to monitor blood flow dynamics also has the great potential to further target cell implantation and to augment overall cell survival in the stroke cavity.

## Histological evaluation of the implanted neural graft

To further evaluate the destiny of transplanted neurons, host brains were perfused at 14 WPI and underwent cryostat sectioning at 50 μm thickness. Anti-GFP staining allowed for the identification of transplanted neurons, including cell bodies and axons (Fig. 7a–c). The boundary between the host and the stroke cavity (containing the grafted cells) could be easily identified with anti-GFAP staining (Fig. S6a). GFAP+ cells could be seen around the lesion boundary, as well as inside the graft. It is important to note that none of the transplanted neurons showed co-labeling with GFAP (Fig. 7d), confirming that the cells being monitored in live animals were not of astroglial lineage. Since the promoters used in this study for donor neurons

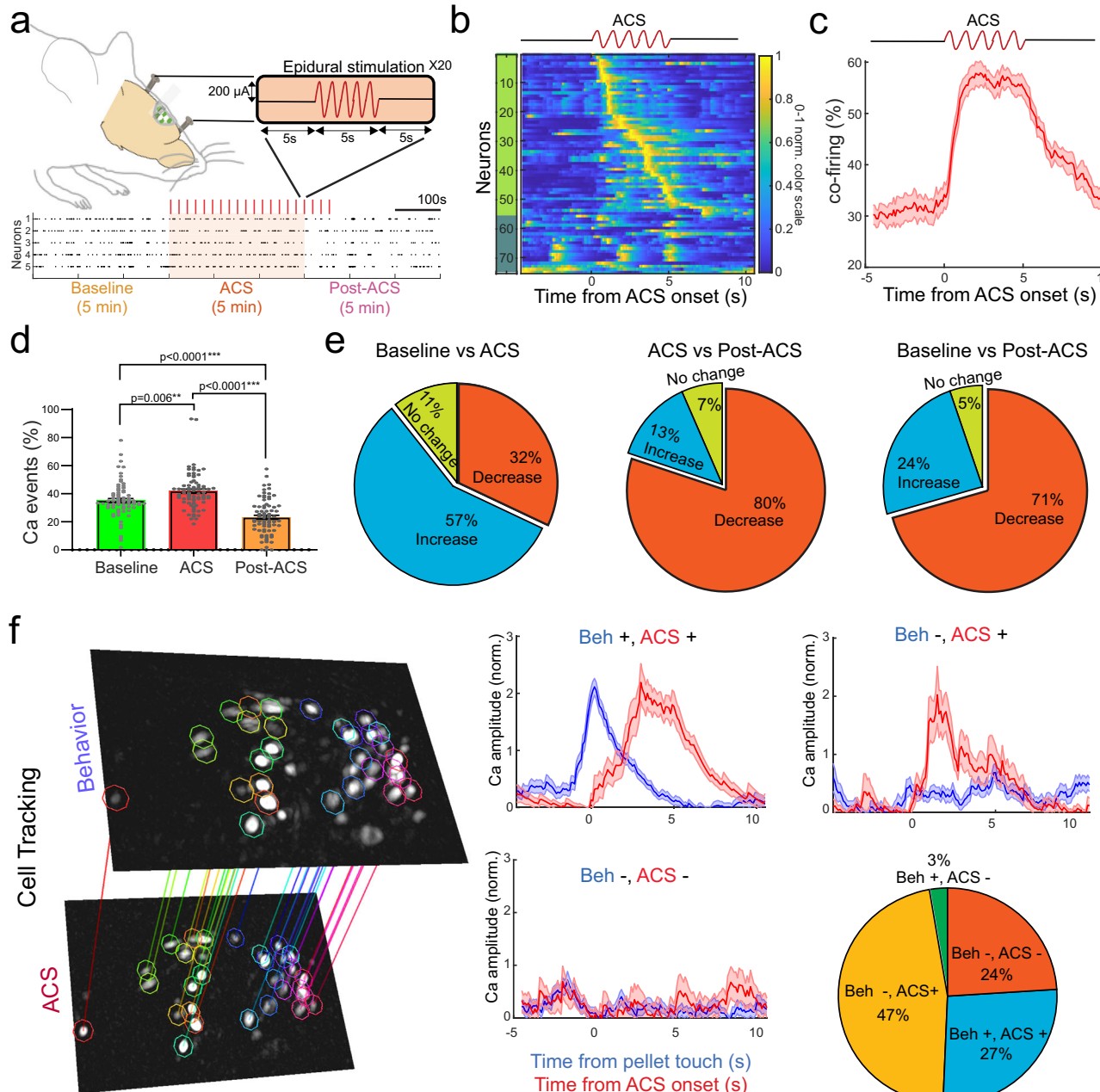

**Fig. 5 | Modulation of neural activity in transplanted neurons using epidural stimulation. a** We manipulated the activity of the transplanted neurons in the entire stroke cavity by applying low frequency alternating current stimulation (ACS) using implanted skull screws and performing calcium imaging for a total of 15 min, including baseline, ACS and post-ACS periods. **b** We then time-locked the calcium traces to ACS onset to determine ACS-modulated units (in green). Plot shows 0-1 normalized color scale. **c** ACS acutely enhanced the neural dynamics of the transplanted network to increase network co-firing or synchrony (trial averaged activity ± SEM). **d** This acute entrainment of neural activity led to a significant increase in the number of detected single events and co-firing during the entire 5 min stimulation period when compared to baseline activity (one tailed paired $t$-test: $**p \leq 0.01$, and $***p \leq 0.001$, $n = 75$ tracked neurons from 6 transplanted mice). All data are presented as the mean ± SEM. Source data are provided as a Source

Data file. **e** At a single cell level, around 57% of transplanted neurons showed an increase in excitability in the transplanted network during ACS, and then followed by a period of reduced neural activity, with ~70% of the neurons showing reduced activity post-ACS compared to baseline. **f** We further probed functional interactions of the transplanted neurons by matching a subset of single neurons from both ACS and behavioral sessions and categorizing them as: behavior (trial averaged activity ± SEM in blue solid line and shaded area) and ACS (trial averaged activity ± SEM in red solid line and shaded area) modulated (Beh+, ACS+), behavior modulated only (Beh+, ACS-), ACS modulated only (Beh-, ACS+), and non-responsive neurons (Beh-, ACS-). Interestingly, around two-thirds of the neurons that were not task-modulated were still responsive to ACS, and almost all task-modulated neurons were responsive to ACS.

are known to drive expression in both excitatory and inhibitory neuronal populations[60,61], we confirmed this expression in transplanted neurons by co-labeling brain sections with anti-GFP, and anti-GAD65/67, a pan-interneuron marker (Fig. S6b). Imaging showed that a subset of GFP+ transplanted neurons expressed

interneuron markers (GFP+, GAD65/67+), while others showed no labeling (GFP+, GAD65/67−), suggesting they were excitatory neurons. This highlights a dynamic interplay between inhibitory and excitatory cell types within the context of transplantation, potentially contributing to the E/I balance (balance of circuit excitation

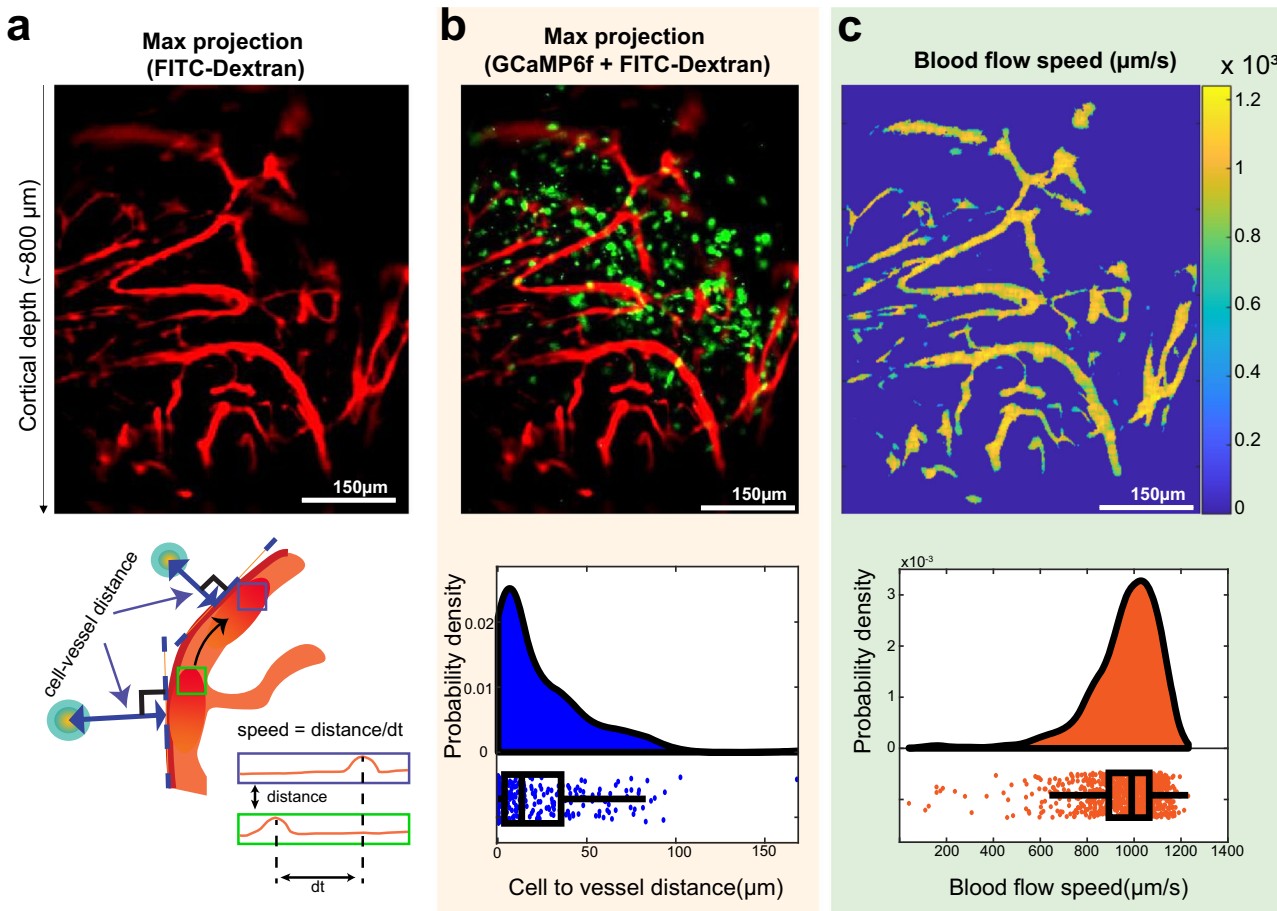

**Fig. 6 | Real time measurement of microcirculatory dynamics within the transplanted graft in vivo. a** Monitoring of blood flow dynamics inside the implanted graft using fluorescein-dextran at 14 WPI. Maximum projection image was pseudo-colored red. **b** Co-registration of calcium imaging and blood flow videos to understand the spatial distribution of surviving transplanted neurons relative to blood vessels. Analysis of distance between neurons and nearest blood vessel revealed that 95% of surviving neurons resided within 76.2 μm of a functional blood vessel (all neurons were within 200 μm). $n = 258$ cells to blood vessel distance from 6 transplanted mice. Source data are provided as a Source Data file. **c** Based on temporal cross-correlation analysis, we determined the average blood flow speed of ~1 mm/s. The histogram displays median and percentile boundary (i.e., 25th and 75th at the box edges) and the whiskers are 1.5 interquartile range. $n = 1000$ randomly selected speed samples from 597618 total samples from 4 transplanted mice. Scale bar: 150 μm. Source data are provided as a Source Data file.

and inhibition) and functional diversity within the transplanted network.

Labeling of host cortical neurons prior to stroke and transplantation revealed substantial infiltration of host axons, labeled with tdTomato, into the GFP+ transplanted graft (Fig. 7e, Fig. S6c). This suggests intricate cellular interactions and dynamic integration of the donor neurons into the host neural network. We also observed that a majority of the transplanted neurons stayed within the stroke cavity at 14 WPI, with sparse migration into the peri-lesional cortex (Fig. S6d). This suggests limited migration capacity of transplanted embryonic neurons in an injured adult brain. This is in sharp contrast to embryonic and early post-natal host brains where transplanted neurons have previously shown to migrate extensively to other cortical regions after injection[62–64]. In addition to the presence of GFP+ axons in the peri-lesional cortex and pre-frontal cortex (Fig. 7a, b), we observed axonal projections from transplanted neurons traversing into the white matter tracts, specifically within the corpus callosum, alongside host axons (Fig. 7c, Fig. S6e). Sub-cortical labeling was evident in lateral septum (Fig. 7c), as well as the ipsilateral striatum (Fig. S6f), showcasing their ability to extend processes into deeper brain regions. Conversely, the contralateral striatum remained void of any observable labeling, emphasizing the specificity of this sub-cortical axonal integration. This intricate interplay between the transplanted and host axons showcases the potential for neural circuitry reorganization and

opens new avenues for understanding the intricacies of brain repair and connectivity restoration.

## Discussion

We report an experimental platform that allows continuous monitoring of transplanted neurons and captures the dynamics of the integration into the host tissue. We utilized an implantable system with optical accessibility that enabled long-term monitoring of activity as neurons undergo highly complex and dynamics changes in connectivity. Our approach mimics natural brain tissue composition including glial and endothelial cells, which provide structural and metabolic support, aiding in neuronal survival and function[65–68]. As a proof of principle, we demonstrated changes in network-wide calcium dynamics and captured the dynamics of cellular function of transplanted neurons throughout the regenerative processes. We also demonstrate reliable modulation of neurons using electrical stimulation and the potential for tracking blood flow dynamics.

Epidural electrical stimulation approaches are relatively less invasive and a promising translation tool for modulating cortical networks. A recent study in non-human primates showed that applying low-frequency ACS improved ensemble co-firing in perilesional cortex and improved motor execution[12]. One key advantage is the ability to modulate neural activity covering large cortical areas, which is essential when transplanting neurons in large stroke cavities typically seen in

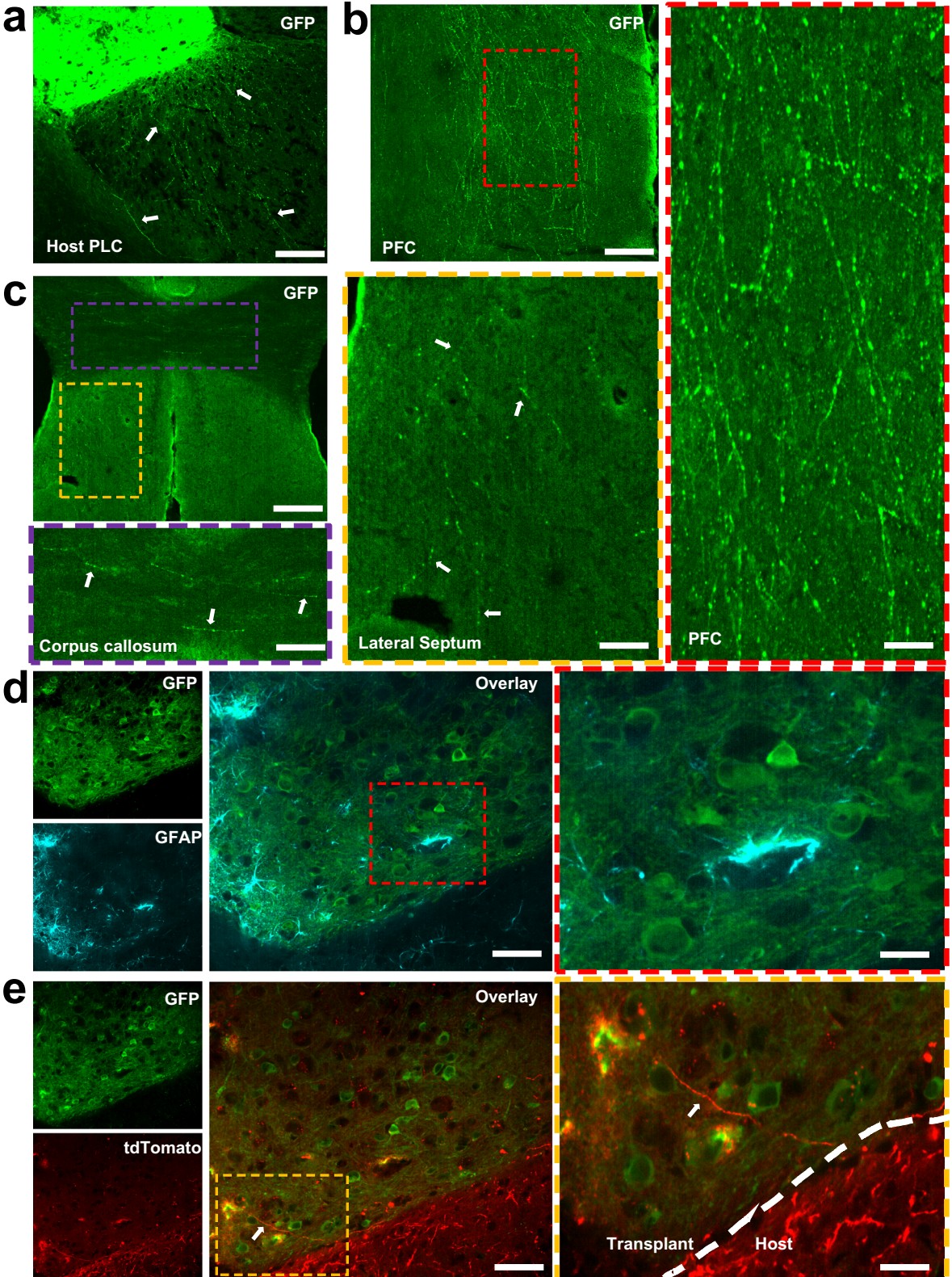

**Fig. 7 | Histological evaluation of the transplanted graft. a** To assess transplanted neuron fate, host brains were perfused and underwent histological analysis at 14 WPI. Anti-GFP staining allowed for the identification of transplanted neurons and their projections into the host peri-lesional cortex (PLC). Scale bar: 100 μm. **b** Large number of axonal projections were also found in pre-frontal cortex (PFC). Scale bar: 100 μm. Inlet: 20 μm. **c** In addition to the presence of GFP+ axons in the cortex, we observed axonal projections from transplanted neurons within the white matter tracts, as well as in lateral septum. Scale bar: 100 μm. Inlets: 50 μm. **d** GFAP+ astrocytes could be seen around the lesion boundary, as well as inside the graft. Scale bar: 75 μm. Inlet: 20 μm. **e** Labeling of host cortical neurons prior to stroke and transplantation revealed substantial infiltration of host axons, labeled with tdTomato, into the GFP+ transplanted graft. Scale bar: 75 μm. Inlet: 20 μm. Histological images are representative from *n* = 6 transplanted mice.

**Table 1 | Key resources table**

|  | Name | Additional information |
|---|---|---|
| **Mice strains** | C57BL/6J | Jax #000664 |
|  | B6.Cg-Tg(Syn1-cre)671jxm | Jax #003966 |
|  | B6.Cg-Tg(Camk2a-cre)T29-1Stl | Jax #005359 |
|  | C57BL/6J-Tg(Thy1-GCaMP6f)GP5.17Dkim | Jax #025393 |
| **Software and algorithms** | MATLAB | Mathworks (https://www.mathworks.com/products/matlab.html) |
|  | CalmAn | Flatiron Institute (https://github.com/flatironinstitut) |
| **Imaging** | nVoke, Proview PRISM lenses | Inscopix (https://www.inscopix.com) |
| **AAV** | hSyn1-tdTomato-WPRE | Addgene #51506 |
| **Antibodies** | GFP | Abcam #ab1218 |
|  | GFAP | Abcam #ab7260 |
|  | GAD65/67 | Abcam #ab183999 |

human patients. In mice with the stroke cavity covering the entire primary motor cortex, ACS reliably modulated ~75% of the transplanted neurons. Within the ACS-modulated population, around two-thirds were movement non-responsive neurons. An important topic for future studies is to understand if long-term open vs. closed-loop stimulation paradigms can help accelerate transplanted circuit integration by increasing task-responsive neurons.

Besides modulating the implanted cells, in vivo visualization of vascular remodeling within the transplanted circuit may aid in identifying the role of various growth factors and their doses needed to achieve sufficient vascularization to prevent cell death, especially in the core of stroke cavity. For example, VEGF is a commonly used to stimulate angiogenesis[69–71], however, the appropriate dosing required to vascularize a large stroke cavity is unclear. Using this platform, the effects of growth factors and their doses can be measured in real time and how they affect neural survival and integration.

While we imaged the activity from both excitatory and inhibitory cortical neurons using pan-neuronal cre lines, this platform can also be used to monitor cell-type specific neural dynamics underlying circuit integration. For example, transplanting embryonic neurons derived from a cross between a GCaMP reporter line (such as Ai148) and a PV-cre (Parvalbumin specific) or GAD2-cre (a pan-interneuron marker) line will allow us to understand how inhibition evolves in a transplanted circuit. Additionally, this platform allows us to adjust the proportions of excitatory and inhibitory neurons within the transplanted circuit and understand its effects not only on the local network but also its functional connectivity with host brain. This can be achieved by using fluorescence-activated cell sorting (FACS) to isolate distinct cell types expressing calcium indicators for transplantation experiments (Fig. S5).

Here, our focus was on long-term observation of calcium related neural dynamics in transplanted neurons in the stroke cavity. While we did observe increased activation of neurons with long-term task training and there was a correlation between activity and behavior, we are unsure about the causal relationship between our observed changes in neural dynamics and improvements in behavior. Future work can use this approach to causally test the role of transplantation (and network integration) in driving motor recovery and further explore cell-type specific neural dynamics. We are optimistic that this will likely shed light on how transplanted network dynamics evolve and interact with host circuitry. There are also notable limitations regarding the translation of this work. Our approach involved early aspiration of tissue after stroke induction; this is not practical. However, it is worth noting that clinical lesions result in extensive encephalomalacia and fluid filled cavities. Future work will need to determine optimal time of implantation and monitoring. Overall, we are hopeful that this platform will accelerate new discoveries that can inform in vivo regeneration of neural circuits.

## Methods
### Animals
Experiments were approved by the Institutional Animal Care and Use Committee at the San Francisco VA Medical Center (Protocol #19-030). Mice strains used in this study are listed in Table 1. We used six wildtype C57BL/6J (Jackson Laboratory, Jax #000664) 10-12 weeks old adult male mice as hosts for stroke and transplantation experiments. For isolating donor embryonic cortical neurons expressing GCaMP6f, we crossed 10-12 weeks old female Ai148D GCaMP6f reporter mice (Jax #030328) with pan-neuronal[60,61], Syn1-Cre (Jax #003966, $n = 3$) and CaMKIIα-Cre (Jax #005359, $n = 3$) male mice. Following calcium binding (i.e., neuronal activation), neurons expressing Cre exhibit a robust increase in EGFP fluorescence. While we used male host mice for current experiments as a proof-of-principle to show dynamic integration of transplanted neurons, future experiments will involve mice from both sex. We also used male 10–12 weeks old Thy1-GCaMP6f (Jax #025393, $n = 3$) as control mice to study network dynamics in healthy cortical networks. The animals were group housed by cohort, including the recovery period after surgery. Animals were kept under controlled temperature (65-75° F) and humidity (40–60%) with a 12-h light/dark cycle; lights on at 06:00 a.m.

### Isolation of donor GCaMP+ embryonic cortical neurons
Since neurogenesis in the rodent cerebral cortex peaks around embryonic day 15 (E15), and cortical astrocytes are initially detected around E16 or E17 due to separate timing of neurogenesis and gliogenesis in the CNS[72], we chose E18 mouse neocortex as the source of donor cells for transplantation. During E18, the pregnant mice were euthanized under 5% isoflurane to isolate and detect fetal cortices expressing GCaMP6f (using Xite fluorescence flashlight, NightSea) (Fig. 1a). The GFP+ cortices were then dissociated by papain digestion and trituration to obtain single cell suspension of cortical neurons expressing GCaMP6f as well as unlabeled glia and vascular endothelial cells. Note: the isolated embryonic cortices contained the entire cerebral cortex, including cells outside of motor cortex.

### M1 photothrombotic stroke
Focal stroke of the entire primary motor cortex (M1) contralateral to the dominant forepaw was be performed using LED illumination of 20 mg/kg Rose Bengal dye under isoflurane (1–2%) and body temperature maintained at 37 C with a heating pad (Fig. 1b). A 2.5 mm × 1.5 mm LED (Digi-Key Electronics) was placed over the center of the M1 region (1.5 mm lateral and 0.5 mm anterior to bregma). Five minutes after Rose Bengal (20 mg/kg) i.p. injection, LED was illuminated (0.125 mA) for 20 minutes, resulting in formation of stroke cavity volume ~3.5 μL in primary motor cortex.

### Cell injection and GRIN lens implantation

Around 60 mins after stroke, burr hole craniotomies were performed for 2 skull screws to allow for secure fixation of the Gradient Refractive Index (GRIN) lens and integrated baseplate (Proview integrated prism lens; Inscopix, Palo Alto, CA). A craniotomy was then performed over the stroke region and the stroke tissue was then aspirated to create a permanent lesion and space for transplanted cells. Using lesion aspiration, we generated consistent sized lesions across all mice. Please note that our previous work has shown that aspiration of stroke cavity is important for a homogenous distribution of the implanted hydrogel, as well as complete gelation and retention within the stroke cavity[34,73]. Embryonic cells were counted and resuspended in 10 mg/ml extracellular matrix (ECM) hydrogel (Matrigel, Corning, #356234) to yield a final concentration of 125k cells/μL, and 3.5 μL of the cell suspension was then injected into the stroke cavity with a micropipette. To deposit cells uniformly within the stroke cavity, Matrigel was used as a scaffold since it is the most widely used ECM for supporting cells and organoids to form 3D structures[74–76], and contains many unique proteins that regulate cell adhesion, growth and differentiation[77]. After cell injection, a 1mmx1mm PRISM lens was implanted into the stroke cavity to image the injected neurons. After insertion of the lens, the lens/baseplate was fixed to the skull using a combination of Metabond and dental cement. No immunosuppressive drugs were given to the mice. Buprenorphine (0.02 mg/kg) was administered for post-operative analgesia and weight carefully monitored for five days following surgery.

### Daily motor training

Two weeks after stroke and cell implantation, mice were habituated to an automated plexiglass behavioral box for 2–3 days (Fig. 1c). Following habituation, they were food restricted and performed daily motor training in the automated behavioral box. The box was controlled by an Arduino microcontroller and custom Matlab scripts, requiring minimal user intervention[44]. At the onset of each trial, an auditory tone was played, and a small window was opened, allowing animals to extend their paw to reach for a small food reward pellet weighing 10 mg. Body weight was measured daily to ensure that it did not drop by more than 10%. Each session consisted of 75 trials, regardless of the number of trials in which the mouse attempted a reach. In our automated systems, these sessions typically took around 30 minutes. Reaching behavior was captured with a lateral camera which acquired images at 200 Hz. Flashes of an LED in the field of view were used to synchronize reaching behavior with the task (door opening, pellet touch, and door close) and neural data. Success/failure and first reach onset time for each trial was manually marked for each behavioral video. For healthy thy1 mice, motor training consisted of daily training for 2 weeks. In our experience, healthy mice display stable motor performance and latent factors within 7 days of training that are preserved long-term.

### Calcium imaging and analysis

At the beginning of imaging sessions, a small head-mounted microscope was attached to the previously implanted prism lens and baseplate (nVoke imaging systems; Inscopix, Palo Alto, CA). Prior to the first imaging session (Imaging FOV: 600 μm x 950 μm), light intensity and gain were adjusted to the minimal settings that could allow for identification of fluorescence transients to minimize photobleaching. After these parameters were established, same settings (LED power and Gain) were used for the remainder of imaging sessions for that animal. Continuous calcium imaging data was acquired using the miniscope at 10 Hz for ~3 months ($n = 6$), and neural data was synced to behavior *post-hoc* using timing pulses generated by a microcontroller. In contrast to head fix imaging using coverslip, we did not observe any changes in the turbidity or scattering of the transplant over the weeks. Images were exported as TIFF stacks and fluorescence signals were extracted using CaImAn imaging data analysis software. After curation of data exported from CaImAn software, custom python scripts were used to evaluate each putative neuron identified by CaImAn software. The denoised calcium traces and the residual traces were calculated by caiman denoising function using deconvolution algorithm (foopsi algorithm[78]). The traces were segmented into each trial, and then baseline correction was applied as following

$$baseline\ corrected = \frac{x_{tr}^{c}(t)}{b_{tr}^{c}} \tag{1}$$

where $x_{tr}^{c}(t)$ is denoised calcium traces c cell and tr trial. $b_{tr}^{c}$ is the averaged S.D of residual traces in −5s to −4s from pellet touch. In Fig. 4d, the single trial traces were displayed based on the baseline corrected df/f. We averaged baseline corrected df/f over trials. To reduce the effect of cell to cell variance, we normalized the each cell activity by using min-max normalization (Fig. 2c, d, and Fig. 5b).

To perform task and cue modulation, the trial data were down sized with each 0.5 s bin. In each bin, the baseline corrected data in 0.5 s were averaged (i.e., 5 10 Hz baseline corrected data were averaged) and then 5 bins from −2.5 s to 2.5 s from pellet touch and −1s to 1 s from door open cue data were used for task and cue modulation calculation. The modulation was performed based on one-way ANOVA ('anova1' with 'multicompare' function in MATLAB) for the multiple time point with bins with multi trial data. The significance was computed for each pair in the data time point by using 'multicompare' function in MATLAB. We consider significance level with $p < 0.05$ for modulation threshold. As the result cells having consistent activity over multiple trials showed significance in the ANOVA modulation test.

For co-firing analysis, post processed calcium transients are converted into binary data. Based on threshold (i.e., standard deviation), the data was converted to 1 for over and 0 for below threshold. Co-firing is calculated dot product of a pair of cells. For FA and principal component analysis (PCA), calcium transient data was extracted from task-related time window (−2.5 s to +2.5 s from pellet-touch time). We performed PCA and calculated explained variance of the components. Based on the explained variance of each dimensional space, number of major dimension (over 90% of total variance in late) was decided. We utilized FA with the number of common shared factor from PCA. As the result of factor analysis shared factor and private factor were decomposed. the shared to total variance was calculated as

$$SOT = \frac{\sum shared}{\sum shared + \sum private} \tag{2}$$

Where $\sum shared$ is shared variance and $\sum private$ is private variance. Calcium transient data in the task related time window was applied to PCA. In order to prevent overfitting, we trained PCA with leave one out of all trial data and performed component calculation by projecting non-trained trial to the trained principal component (PC) space. For long-term cell tracking, we mapped single neuron activity across multiple training sessions using CaImAn cell registration tools and verified the results by manual inspection[79].

### Epidural stimulation and analysis

Prior to injecting neurons in the stroke cavity, we implanted cranial screws around the stroke cavity to generate low-frequency electrical stimulation (1 Hz, $n = 3$ mice). Sessions with epidural stimulation were performed around 14 WPI following behavioral sessions to allow reliable tracking of neurons across behavior and stimulation sessions. Each recording and stimulation session lasted 15 min and consisted of three 5 min blocks: baseline, stimulation, and post-stimulation periods. Sinusoidal waveform with 200 μA current (5 s ON, 10 s OFF) was passed through the skull screws while recording the calcium activity (at 10 Hz) of the transplanted neurons for a total of 20 repetitions per mouse.

To calculate neurons that were modulated by stimulation, we used ANOVA modulation (the same modulation test for task related modulation test) having 1 s time bin with −5s from the simulation onset to stimulation end (+5 s). In addition, to test 5 s continuous simulation modulation, we applied amplitude difference-based modulation test. 95 percentiles of peak amplitude in trials of task related calcium transients are trial averaged. The trial averaged calcium transients are compared to shuffled data. The value above the shuffled data was determined as ACS-modulated neurons.

Calcium event was measured by a deconvolution method using constrained foopsi algorithm[78], which is implanted in CaImAn toolbox. We counted events during 5 min baseline (before ACS start), 5 min ACS (with repeated 5 s stim and 10 s inter-stimulation interval), and 5 min post-ACS periods. The event count was normalized by total event count for 15 m recording (including baseline, ACS, and post-ACS) for each cell. This event ratio for each cell were classified into 'increased', 'decreased', and 'no-change' that shows 10% event ratio increase, 10% event ratio decrease, and less than 10% event ratio change respectively.

### Imaging and analysis of microcirculatory dynamics

The one-photon miniscope has been previously used to perform in vivo blood flow imaging in healthy rodent and macaque brains[59,80]. 1–2 days after stimulation experiments, mice were injected i.p. with 0.2–0.25 ml fluorescein-dextran (Sigma, FD2000S, 10 mg/ml, $n = 4$ mice) to label blood plasma and identify blood vessels in the implanted neural graft. 1 hour after the i.p. injection, videos were acquired using the miniscope at 60 Hz for 5 mins. We determined erythrocyte flow speeds using a method based on temporal cross-correlations. We first motion corrected the acquired videos, and then acquired a maximum projection image from each mouse. This highlighted blood vessels, which due to blood flow exhibited greater temporal fluctuations in intensity than background pixels. Further image processing was performed after the max projection to detect shape of vasculature. $2.2\,\mu m^2$ spatial box moving filter smoothen the outer lines of the vasculature and increase the contrast to the background. After that spatial filtering with a threshold which best remove the background was applied. We then measure the distance between cells to the vasculature and blood flow speed.

Cell distance to the vasculature was calculated by searching the vasculature that has minimum distance to a cell. For the blood flow speed calculation, we made an $8.8\,\mu m^2$ spatial box floating on the vasculature. We computed every pixel time difference from the maximum cross-correlation of every two pixels within the spatial box and found the maximum speed in the box. To compute the speed of erythrocytes passing through a specific reference pixel, we computed the cross-correlation between the intensities of the reference pixel and neighboring pixels. For this calculation we used 30 s segments of data. We computed the centroid of each of the cross-correlograms and then determined the speed of erythrocytes passing through the reference pixel as the average of the distances divided by the time between two image frames.

### Statistical analysis

Figures show mean ± standard error of the mean (s.e.m.). Parametric statistics were performed by using GraphPad Prism (ver. 9, GraphPad software). One-way ANOVA was performed for more than three conditions in a group (i.e., early, mid, and late condition). Paired one-tailed $t$-test was used to measure the significance of the differences between two conditions (i.e., early-mid or mid-late); ns $P > 0.05$, *$P \leq 0.05$, **$P \leq 0.01$, and ***$P \leq 0.001$ denote statistical significance.

Correlation coefficient was performed by using LME with random intercept and random slope model (using MATLAB "fitlme") for correlation between behavior performance and task modulation in Fig. 2f and for correlation between task attempt ratio and task modulation in

Supplementary Fig. 1f. Correlation coefficient 'r' was calculated by slope of fixed effect. For random effect, slope 'random r-std' and intercept 'random intercept-std' were calculated.

Histogram was calculated by using 'Raincloud plots'[81] for Fig. 4f, and Fig. 6b, c. Median, 25th, and 95th percentile were calculated and displayed in the boxplot. Whiskers connected to the boxplot shows data greater than $q3 + 1.5 \times (q3-q1)$ or less than $q1-1.5 \times (q3-q1)$, where q1 and q3 are the 25th, and 75th percentile.

### Immunohistochemistry

For histological evaluation of host axons innervating into the transplanted graft, three host mice were injected with AAV5-hSyn1-tdTomato-WPRE in the brain tissue surrounding primary motor cortex 4 weeks prior to stroke and implantation surgery (0.5 µl each at coordinates in mm relative to Bregma: AP: 0.5, ML: 0.5, DV: 0.5 and AP: 0.5, ML: 1.5, DV: 0.5). Mice were transcardially perfused at 14 weeks post-implantation with 0.9% saline followed by 4% paraformaldehyde (PFA, in 0.2 M PBS) to fix brain tissue prior to its removal from the skull. Brains were post-fixed in 4% PFA for 24 h then cryopreserved in 20% sucrose with sodium azide (Sigma) at 4 C. Using a cryostat (Leica), 50 µm thick histological sections were cut directly onto microscopic slides. Brain sections were washed 3 × 5 min with 0.01 M PBS, followed by 1 h of blocking and permeabilization with 5% normal goat serum (NGS, Sigma) and 0.3% Triton X-100, diluted in PBS. Primary antibodies against green fluorescent protein (anti-GFP, 1:1000, ab13970, Abcam), glial fibrillary acidic protein (anti-GFAP, 1:1000, Abcam), and glutamic acid decarboxylase (anti-Gad65/67, 1:300, ab183999, Abcam) were applied, diluted in PBS + 0.3% Triton + 5% NGS, and incubated at 4 C overnight. After rinsing off the primary antibodies (3 ×5 min PBS), secondary AlexaFluor antibodies (1:500, Life Technologies) were applied for 1 h at room temperature followed by 3 × 5 min washed with PBS. Finally, sections were cover slipped with Vectashield mounting medium and stored at 4 C prior to imaging with SoRA spinning disk confocal microscope (Fig. 7), as well as a Zeiss fluorescence microscope (Fig. S6).

### Reporting summary

Further information on research design is available in the Nature Portfolio Reporting Summary linked to this article.

## Data availability

Source data are provided as a Source Data file. The calcium imaging data at Early, Mid, and Late used in this study have been deposited at https://zenodo.org/records/10003677 (https://doi.org/10.5281/zenodo.10003676). Source data are provided with this paper.

## Code availability

Custom MATLAB codes are available at https://github.com/gangulylab/CellTPCode.

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

## Acknowledgements

This work was supported by awards from the Department of Veterans Affairs (I01 5I01RX001640 to KG) and Weill Neurohub Institute to KG. This was also supported by the American Heart Association (20POST35200183 to HG).

## Author contributions

H.G. and K.G. were responsible for study conception and design. H.G. performed embryonic cortical cell isolation and implantation surgeries. S.B. and H.G. performed stroke surgeries and behavioral experiments. H.G. performed cell tracking, blood flow and stimulation experiments. H.G. and S.B. conducted immunohistochemistry and microscopy. K.K., H.G., and K.G. contributed to data analysis, interpretation, and drafting figures. H.G. and K.G. drafted the manuscript. All authors revised the manuscript and approved the final version.

## Competing interests

The authors have no competing interests to declare.
