## [Peer Review File · Nature Communications]

Emergence of task-related spatiotemporal population dynamics in transplanted neuronsREVIEWER COMMENTS

Reviewer #1 (Remarks to the Author):

This is a very interesting and relevant study performed at a high methodological level with modern and advanced technology. It advances our knowledge and possibilities to follow transplanted neuron activity and link it with the defined behavior. The work is original and well-performed. The data support the conclusions.

The main weakness of the study is that the authors do not perform any morphological or histological analysis of the transplant postmortem to get a better insight into the type of neurons which they are monitoring in live animals.

Reviewer #2 (Remarks to the Author):

The current study conducts chronic calcium imaging using a head-mounted microscope to identify the neuronal network dynamics of transplanted neurons. The study reveals that the transplanted cell networks acquire task-related population activity patterns that resemble healthy networks. The factor analysis also demonstrates that transplanted cell networks gain task-related co-firing and rotational neuronal dynamics, as seen in healthy cortical networks. The study further validates multiple modalities, including cell tracking, alternating current stimulation, and blood flow dynamics. The authors' approach is promising for the study of mechanisms underlying neuronal network reorganization after transplantation and to validate recovery determinants, such as post-stroke intervention (rehab and stimulation therapies) and angiogenesis. The analysis methods for network dynamics are well established in previous research conducted by the authors' group for some time. There are a few modest criticisms.

The study did not show how behavioral performance recovered after the stroke in a time-dependent manner. The authors should present a figure which shows the behavioral performance along the timeline. They preferably use early, middle, and late time blocks, but weekly or daily data presentations would be better for this.

Does the activity pattern change/emerge because of repeating skilled reach-to-grasp task training, or does it occur spontaneously by integration after transplantation?

The article will be more insightful if the authors provide data from animals without daily rehabilitative training but intermittent reach-to-grasp test (e.g., weekly). If not, the correlation analysis of the reaching attempt numbers and the network dynamics may help to interpret this aspect.

In Figures 1c & d, 4d, and 5b, the color scale units in the heat map of neuronal activation are not displayed.

Figure 2f shows the correlation between the task success rate and the task modulation proportion. Since both measurements naturally increase with time, analyzing the correlation using a linear mixed effect model with time points as a random effect may be more informative.

The study used factor analysis (FA) to determine the shared-over-total variance (SOT). Line 128 indicates, "Our SOT measure used the first two factors as our analysis revealed that the top two factors explained ~90% of the neural variance (Fig. S2).", but Figure S2 shows the principal component analysis (PCA) data. It is understood that FA and PCA are different as components of PCA are always orthogonal. The factors in FA are not required as orthogonal, which means the factors may account for overlapping variance. Please explain how FA and PCA are used differently for SOT analysis.

Figure 3c shows trial-averaged projections of network activity. Does it show a representative animal? If so, how similar trends did the other animals show? Please explain.

Although the neuron-specific promoters sorted the transplanted cells, it is preferable to perform immunohistochemical analysis with the post-mortem brain tissues to confirm cellular characteristics (e.g. NeuN, GFAP, CaMKII, and GAD). Similarly, histological confirmation of the axonal infiltration and synapse formation into the graft will convince the finding in the study.

Reviewer #3 (Remarks to the Author):

In the paper by Ghuman et al., the authors describe an interesting approach for monitoring and manipulating transplanted embryonic cortical neurons *in vivo*. Thanks to the proposed method, they could visualize the emergence of a network of donor neurons that progressively tune their activity to the voluntary movement (reach-to-grasp) in the weeks that follow a cortical injury. They demonstrate the feasibility of activating these cells by epidural stimulation with ACS. Finally, they show the presence of blood vessels in the graft on the last week of training and found that most neurons lay in proximity to blood vessels. The technical approach is appealing and the results of the task-modulated activation of the transplant are novel and intriguing. Previous work, cited in the paper, longitudinally monitored *in vivo* the activity of neuronal cells within a graft (Mansour 2018), but none showed temporal locking of the activity of the grafted network to a voluntary (reach-to-grasp) movement, a feature that is consolidated along the weeks of rehabilitative training. As written in the introduction, the approach proposed here will allow for greater insights into the integration of cell transplants into injured networks, however, in the present form some of the key results (including the epidural stimulation) are proof-of-principle or lack crucial control groups.

All things considered, the approach and the results could be of interest to a wide audience within the neuroscience and neurology communities, but the paper has some major flaws that need to be corrected.

Major concerns:

1) The procedure is rather complex, with multiple surgical interventions, albeit in a single session. For clarity reasons, the necessity for aspiration of the tissue should be spelled out. Importantly, the lesion caused by tissue aspiration is not necessarily comparable to a real ischemic lesion and only some of the inflammatory cascades induced by photothrombosis might be activated one hour after injury. This issue strongly reduces the relevance of the findings and the translatability of the findings to clinical settings.

The authors should make a control group to verify if cell synchronization of the transplanted cells is occurring even without tissue aspiration. Possibly, they should also wait a few days from the stroke to perform the host cells transplantation.

2) My major concern is the actual amelioration of the motor capacities in stroke mice induced by the injected cells. Indeed, we cannot be sure that the donor cells are actually effective in ameliorating motor recovery since behavioral analysis is lacking. There is only one graph that correlates the task success rate with task modulation but no results whatsoever on the progressive recovery of reach-to-grasp are presented. The discussion mentions that “we are not sufficiently powered to determine how the transplanted neurons contributed to recovery”, however, this is a very important point. The authors should show how the task success rate changes over the weeks of training and compare these to control groups injected with medium or saline. Also, they should evaluate by performing additional motor tests if the progression is task-specific or generalized to other motor functionalities.

3) Along these lines, another control group that is missing is the injection of the medium ECM without the cells. The growth factors in the medium could be responsible for increasing the plasticity of the spared neuronal networks, improving their task-locked activity, and ultimately for behavioral recovery.

4) Important information on the Thy1-GCaMP control is missing. Specifically, I could not find any reference in the results nor in the methods sections on the week(s) of training. Are the data shown for Thy1 mice trained for the same number of weeks as transplanted mice? Are they naive?

5) Two promoters are used to drive cell-specific expression of GCaMP6f, Syn, and CaMKII α . As far as I know, the second promoter does not drive pan-neuronal expression but is specific to excitatory neurons. Thus, the authors should describe if the findings are equivalent for Syn and CaMKII. Possibly, they could show by FACS analysis the proportion of excitatory and inhibitory neurons selectively under each promoter. This is relevant also for the comparison with the Thy1-GCaMP control groups, where only excitatory neurons are labeled.

Accordingly, please revise lines 253-254.

6) Figure S1 may suggest that the thickness of the transplant changes during the weeks. The authors should quantify this variation at least at one additional time point during the early phase of training. Does the thickness of the tissue where active cells are present change during the weeks? Are they evenly distributed along the volume, or do you ever see cell layers formed during the weeks? Do you see any dependence on the depth from the surface/proximity to the donor tissue of the task-activated cells in the early compared to the late phase of training?

7) It is not clear whether the newly formed blood vessels are “functional”, as claimed by the authors. For instance, they could be not fully mature and leaking; I suggest the authors could evaluate the permeability of the vessels to confirm their functionality. Also, it would be interesting to evaluate if blood vessel density correlates with behavioral recovery.

8) Since the approach presented here allow for a longitudinal evaluation of blood vessels, it would be interesting for the reader to have a timeline of their formation: are they already formed two weeks after stroke? Do they form during the training weeks? Does it correlate with behavioral recovery?

Minor points:

1) I could not find the total number of neurons that are (or are not) task modulated, only relative values (%). Please add this information.

2) The legend at the end of Figure S1 states that the host tissue surrounding the transplant consists of the pre-motor cortex. This is true for just one side. Please correct.

3) Figure 3, panel C: for clarity, please use a different color for healthy and early training groups.

4) Lines 195-6: “This allowed us...based on their functional interaction with the perilesional cortex”. This claim is a guess, but it is not supported by the data.

5) Please add average and error (SD or SEM) to quantify the distance of the cells to the blood vessels.

6) Video 1 could be improved by adding a reference to the time point at which the reaching started. Also, the cells with task-related activity should be evidenced by a ROI and the relative traces of activation could be shown on the side of the video.

7) Video 4: please show the cells as they appear along the depth together with the blood vessels.

8) Please comment on the fact that the expression of GCaMP could change over time, with more positive cells over the weeks. Do you see an increased fluorescence over the weeks?

9) In the discussion, please comment on the importance of glial and endothelial cells in the transplant. Would the results change if only neurons were transplanted?

10) Does the turbidity/scattering of the transplant change over the weeks? Please add this information in the methods section.

Reviewer #4 (Remarks to the Author):

Ghuman et al presented a toolbox for long-term tracking and monitoring of transplanted embryonic neurons in adult mice after cortical injury. Using this platform, they demonstrated that transplanted neurons were capable to integrate into the damaged cortical area and formed healthy networks. The results look very promising but I have several questions listed below:

1. I have some general questions about the destiny of transplanted embryonic cortical cells. Were GCaMP6f+ transplanted neurons developed into both excitatory neurons and inhibitory neurons? What was the survival rate after transplantation? Were the total survival neurons reduced with time? Can transplanted neurons migrate into other brain areas?
2. The authors indicated that adult host mice were transplanted with embryonic cortical cells from either Syn1-GCaMP6f or CaMKII-GCaMP6f mice. It seems that the authors did not further distinguish these two population for all of their analysis. I am wondering why. Cortical cells from Syn1-GCaMP6f mice might have more inhibitory neurons being labeled with GCaMP6f than that of CaMKII-GCaMP6f mice. This might affect the calculation for % of task modulation neurons.
3. Were panels (E, M, L) in Figure 2C from the same mouse? If so, I am wondering why the cell maps changed so dramatically. If not (i.e., the panels were from three different mice), the authors should show the representative images from the same mouse to demonstrate increase of task-modulated neurons with time.
4. Fig 4e should include a health control (H) for comparison (as in Figure 2e and Figure 3b, 3c).
5. Figure 2e, figure legend indicated "n = 6 transplanted mice, n= 3 neurotypical mice". Are there only a total of 6 transplanted mice being measured at three different time points (E,

M, L)? The n number seems too low.

6. What is R square for Figure 2f?

General

We would like to thank the reviewers of our manuscript, “Emergence of task-related spatiotemporal population dynamics in transplanted neurons”. Since receiving the reviewer’s valuable feedback, we have made significant changes to our manuscript.

Specifically, we appreciate the general guidance about the scope of our manuscript. As we described in the introduction, the main goal of this manuscript is a proof-of-principle demonstration of long-term monitoring of transplanted neurons in the stroke cavity. We also demonstrate proof-of-principle for modulation of transplanted neurons. We are optimistic that this method will then allow us and others to ask fundamental questions about how transplantation and restoration of neural dynamics might causally drive recovery. We have also included a limitations section to address these points.

We are also grateful that all reviewers appreciate the general advance of our method. We appreciate comments such as:

- *“This is a very interesting and relevant study performed at a high methodological level with modern and advanced technology...work is original and well-performed. The data support the conclusions.”*
- *“...promising for the study of mechanisms underlying neuronal network reorganization after transplantation and to validate recovery determinants.”*
- *“The technical approach is appealing and the results of the task-modulated activation of the transplant are novel and intriguing.”*

We are also hopeful that our revised manuscript provides a significant advance in the field of neural transplantation and will be of interest to the field. Below we address specific concerns and comments.

Reviewer #1 (Remarks to the Author): This is a very interesting and relevant study performed at a high methodological level with modern and advanced technology. It advances our knowledge and possibilities to follow transplanted neuron activity and link it with the defined behavior. The work is original and well-performed. The data support the conclusions.

We greatly appreciate this comment!

The main weakness of the study is that the authors do not perform any morphological or histological analysis of the transplant postmortem to get a better insight into the type of neurons which they are monitoring in live animals.

We thank the reviewer for their comment regarding missing postmortem histological analysis of the implanted neural grafts. We have now incorporated Figure 7 into our manuscript to include histological data. We show that the transplanted cells being monitored in live animals are not of astroglial lineage (GFP+, GFAP-), express markers for both inhibitory neurons (GFP+, GAD65/67+) and excitatory neurons (GFP+, GAD65/67-), and extend their axons into host peri-lesional and sub-cortical areas.

We also provide new data with both labeling of host neurons prior to stroke and neural transplantation as prior; this revealed significant infiltration of host axons into the transplanted neural graft. Also, we noticed that majority of the transplanted neurons stayed within the stroke cavity after 3 months, with sparse migration into the peri-lesional cortex.

Reviewer #2 (Remarks to the Author): The current study conducts chronic calcium imaging using a head-mounted microscope to identify the neuronal network dynamics of transplanted neurons. The study reveals that the transplanted cell networks acquire task-related population activity patterns that resemble healthy networks. The factor analysis also demonstrates that transplanted cell networks gain task-related co-firing and rotational neuronal dynamics, as seen in healthy cortical networks. The study further validates multiple modalities, including cell tracking, alternating current stimulation, and blood flow dynamics. The authors' approach is promising for the study of mechanisms underlying neuronal network reorganization after transplantation and to validate recovery determinants, such as post-stroke intervention (rehab and stimulation therapies) and angiogenesis. The analysis methods for network dynamics are well established in previous research conducted by the authors' group for some time. There are a few modest criticisms.

We thank the reviewer for your comments and insightful critique of our manuscript! We have incorporated the reviewer's feedback, addressed the specific concerns outlined below, and present a revised manuscript for your consideration.

The study did not show how behavioral performance recovered after the stroke in a time-dependent manner. The authors should present a figure which shows the behavioral performance along the timeline. They preferably use early, middle, and late time blocks, but weekly or daily data presentations would be better for this.

We appreciate this comment. We have now included the weekly behavioral performance from all mice in Figure S1a. We observed significant increase in task success rate over the course of rehabilitation training (One-way ANOVA: $p = 0.0007$, $R^2 = 0.53$).

Does the activity pattern change/emerge because of repeating skilled reach-to-grasp task training, or does it occur spontaneously by integration after transplantation? The article will be more insightful if the authors provide data from animals without daily rehabilitative training but intermittent reach-to-grasp test (e.g., weekly). If not, the correlation analysis of the reaching attempt numbers and the network dynamics may help to interpret this aspect.

We would like to thank the reviewer for the suggested approach. To answer this question, we tested correlation with linear mixed effect model (LME) between reach attempt rate and task-related neural modulation for all mice. The results show these are moderately correlated (LME: (fixed) $r = 0.65$, $p = 0.01$, $R^2 = 0.29$; (random) intercept-std = 4.6, r-std = 7.1×10^{-11}). This suggests that repetition of a skilled behavior is likely to increase the task-related network dynamics in transplanted neurons. We have now included this analysis in Figure S1e.

In Figures 1c & d, 4d, and 5b, the color scale units in the heat map of neuronal activation are not displayed.

We have now added normalized color scale information for all plots showing heat map of neuronal activation. We also added more details about the color scale, baseline correction and normalization in methods.

Figure 2f shows the correlation between the task success rate and the task modulation proportion. Since both measurements naturally increase with time, analyzing the correlation using a linear mixed effect model with time points as a random effect may be more informative.

We agree with that using linear mixed effect model provides better interpretation of the relationship between behavior performance and task modulation. We calculated the fixed and random effects of LME (slope for correlation coefficient and intercepts). Slope of fixed effect for correlation coefficient (r), and standard deviation of intercept (intercept-std) and slope (r -std) of random effect were measured. We found that behavior performance was moderately correlated with the proportions of task-modulated neurons in each of the transplanted networks (**Fig. 2f**, LME fitting (fixed) r : 0.6, p : 0.002, R^2 : 0.47; (random) intercept std: 6.4, r -std: $4.4e-12$). We have now edited the manuscript to reflect the LME model.

The study used factor analysis (FA) to determine the shared-over-total variance (SOT). Line 128 indicates, “Our SOT measure used the first two factors as our analysis revealed that the top two factors explained ~90% of the neural variance (Fig. S2).”, but Figure S2 shows the principal component analysis (PCA) data. It is understood that FA and PCA are different as components of PCA are always orthogonal. The factors in FA are not required as orthogonal, which means the factors may account for overlapping variance. Please explain how FA and PCA are used differently for SOT analysis.

Thank you for requesting more clarification on this analysis. We utilized PCA to quantify variance of orthogonal dimensions of the data to measure how many dimensional spaces explained most of the variance. We found that the first two components explained ~90% of variance in the data. This indicated that by using two common factors in FA ~ 90% SOT will be calculated in FA. We then used the same number of common factors for FA for co-firing activity. We have now revised the manuscript to make it more clear how PCA and FA are used for SOT analysis.

Figure 3c shows trial-averaged projections of network activity. Does it show a representative animal? If so, how similar trends did the other animals show? Please explain.

We appreciate this comment. To clarify this point, we have now added the neural trajectories for all transplanted mice in Figure S2e. We also added more details in methods on how these plots were made. Indeed, all mice show similar projections of neural activity; noisy rotational dynamics in early sessions that become progressively more similar to each other as well as healthy circuitry.

Although the neuron-specific promoters sorted the transplanted cells, it is preferable to perform immunohistochemical analysis with the post-mortem brain tissues to confirm cellular characteristics (e.g. NeuN, GFAP, CaMKII, and GAD). Similarly, histological confirmation of the axonal infiltration and synapse formation into the graft will convince the finding in the study.

We thank the reviewer for requesting postmortem histological analysis of the implanted grafts. We have now incorporated Figure 7 into our manuscript to include this histological data. We show that the transplanted cells being monitored in live animals are not of astroglial lineage (GFP+, GFAP-), express markers for inhibitory neurons (GFP+, GAD65/67+) and excitatory neurons (GFP+, GAD65/67-), and extend their axons into host peri-lesional and sub-cortical areas. Labeling of host neurons prior to stroke

and neural transplantation revealed significant infiltration of host axons into the transplanted neural graft. Also, we noticed that majority of the transplanted neurons stayed within the stroke cavity after 3 months, with sparse migration into the peri-lesional cortex.

Reviewer #3 (Remarks to the Author): In the paper by Ghuman et al., the authors describe an interesting approach for monitoring and manipulating transplanted embryonic cortical neurons in vivo. Thanks to the proposed method, they could visualize the emergence of a network of donor neurons that progressively tune their activity to the voluntary movement (reach-to-grasp) in the weeks that follow a cortical injury. They demonstrate the feasibility of activating these cells by epidural stimulation with ACS. Finally, they show the presence of blood vessels in the graft on the last week of training and found that most neurons lay in proximity to blood vessels. The technical approach is appealing and the results of the task-modulated activation of the transplant are novel and intriguing. Previous work, cited in the paper, longitudinally monitored in vivo the activity of neuronal cells within a graft (Mansour 2018), but none showed temporal locking of the activity of the grafted network to a voluntary (reach-to-grasp) movement, a feature that is consolidated along the weeks of rehabilitative training. As written in the introduction, the approach proposed here will allow for greater insights into the integration of cell transplants into injured networks, however, in the present form some of the key results (including the epidural stimulation) are proof-of-principle or lack crucial control groups. All things considered, the approach and the results could be of interest to a wide audience within the neuroscience and neurology communities, but the paper has some major flaws that need to be corrected. Major concerns:

1) The procedure is rather complex, with multiple surgical interventions, albeit in a single session. For clarity reasons, the necessity for aspiration of the tissue should be spelled out. Importantly, the lesion caused by tissue aspiration is not necessarily comparable to a real ischemic lesion and only some of the inflammatory cascades induced by photothrombosis might be activated one hour after injury. This issue strongly reduces the relevance of the findings and the translatability of the findings to clinical settings. The authors should make a control group to verify if cell synchronization of the transplanted cells is occurring even without tissue aspiration. Possibly, they should also wait a few days from the stroke to perform the host cells transplantation.

We agree with the reviewer that inflammatory cascades induced by photothrombosis are likely to be different if cell implantation was performed a few days after the stroke. Indeed, multiple studies have shown that a delayed neural cell injection (7-14 days post-stroke) leads to better cell survival, likely due to the reduction in pro-inflammatory cytokines that are released due to stroke. However, as described in the general outline, we aim to provide a proof-of-principle method that transplanted neurons can indeed be tracked long-term in live animals and show task-related dynamics when injected soon after stroke. We are indeed excited that future studies can address fundamental questions about optimal approaches for implantation (particularly from a translational perspective) as well as examine the causal contribution of transplantation and restoration of dynamics.

Regarding tissue aspiration, our previous work has shown that aspiration of stroke cavity is essential for a homogenous coverage of the cavity (Ghuman et al. 2015); poor aspiration leading to incomplete gelation and retention of the ECM hydrogel, as well as affecting its biodegradation properties. Additionally, without lesion aspiration, we are extremely limited with graft volume that can be safely injected into the stroke tissue without causing significant tissue swelling and injection backflow. Implantation of the PRISM lens into non-aspirated tissue will cause further compression and inflammation of the surrounding host tissue. To avoid these undesirable effects, we performed aspiration of the stroke tissue and we have now amended

our manuscript to make this point clearer. We have added to the limitations section at the end of the manuscript.

2) My major concern is the actual amelioration of the motor capacities in stroke mice induced by the injected cells. Indeed, we cannot be sure that the donor cells are actually effective in ameliorating motor recovery since behavioral analysis is lacking. There is only one graph that correlates the task success rate with task modulation but no results whatsoever on the progressive recovery of reach-to-grasp are presented. The discussion mentions that “we are not sufficiently powered to determine how the transplanted neurons contributed to recovery”, however, this is a very important point. The authors should show how the task success rate changes over the weeks of training and compare these to control groups injected with medium or saline. Also, they should evaluate by performing additional motor tests if the progression is task-specific or generalized to other motor functionalities.

We agree that the role of transplanted cells in behavioral recovery is important. We have now included the weekly behavioral task success rate in Figure S1a to show improvements in task performance for all of the transplanted mice. We observed significant increase in task success rate over the course of rehabilitation training (One-way ANOVA: $p = 0.0007$, $R^2 = 0.53$). We would like to reiterate that our claim in this study is not about functional improvements due to transplanted cells, but rather a proof of principle that transplanted neurons can be tracked and manipulated long-term in live animals, and that transplanted neurons are indeed capable of producing dynamics similar to healthy circuitry. This will allow researchers to monitor the evolving dynamics of transplanted neurons and better understand real-time circuit integration, regardless of functional outcomes. As suggested by the editor during our discussions about the scope of our manuscript, we have added a limitations section on the limitations of our approach and lack of any claims about causality.

3) Along these lines, another control group that is missing is the injection of the medium ECM without the cells. The growth factors in the medium could be responsible for increasing the plasticity of the spared neuronal networks, improving their task-locked activity, and ultimately for behavioral recovery.

We agree with the reviewer that ECM alone could be responsible for the behavioral improvements in transplanted mice. This has been previously shown to be the case in our work (Ghuman et al 2018). However as explained above, the current work highlights a proof of principle rather than claiming functional role of implanted cells in improving recovery. We have amended our discussion to reflect the limitation of our current work in terms of claiming functional improvements.

4) Important information on the Thy1-GCaMP control is missing. Specifically, I could not find any reference in the results nor in the methods sections on the week(s) of training. Are the data shown for Thy1 mice trained for the same number of weeks as transplanted mice? Are they naive?

Thank you for this comment and requesting more details on the Thy1 mice. We know that healthy (i.e. non-stroke) mice display stable task-related dynamics long-term after reaching plateau performance, usually within 7 days of motor training. After reaching stable motor performance, naïve mice display stable latent factors that were preserved during the course of our monitoring. For this reason, we used the neural data from Thy1 mice at 2 weeks post-training. We have clarified and expanded the methods to include this information. We have also added additional tracked analysis for Thy1 neurons in Figure S3c,d.

5) Two promoters are used to drive cell-specific expression of GCaMP6f, Syn, and CaMKII α . As far as I know, the second promoter does not drive pan-neuronal expression but is specific to excitatory neurons. Thus, the authors should describe if the findings are equivalent for Syn and CaMKII. Possibly, they could show by FACS analysis the proportion of excitatory and inhibitory neurons selectively under each promoter. This is relevant also for the comparison with the Thy1-GCaMP control groups, where only excitatory neurons are labeled. Accordingly, please revise lines 253-254.

Thank you for clarifying this point. Although CaMKII α promoter was previously thought to be specific for excitatory neurons, recent studies have shown that it does drive expression in certain populations of inhibitory neurons (references in methods). It is also important to point that while the promoters likely allow monitoring somewhat different neural populations, the transplantation procedure was the same and there was no sub-selection. To further evaluate any differences in observed task-related neural dynamics in transplanted mice with Syn and CaMKII α promoter, we measured task modulated populations across both groups and found no significant differences (unpaired t-test: (early) $p = 0.73$, (mid) $p = 0.41$, (late) $p = 0.67$). We have amended our results to reflect this finding.

6) Figure S1 may suggest that the thickness of the transplant changes during the weeks. The authors should quantify this variation at least at one additional time point during the early phase of training. Does the thickness of the tissue where active cells are present change during the weeks? Are they evenly distributed along the volume, or do you ever see cell layers formed during the weeks? Do you see any dependence on the depth from the surface/proximity to the donor tissue of the task-activated cells in the early compared to the late phase of training?

We agree with the reviewer that changes in depth-dependent activation and modulation of transplanted neurons is an important point to be considered. We found that during early phase, although there were more active cells being monitored at a depth of 400-600 μm (which includes layer V in intact mouse cortex), the relative proportions of task-modulated neurons remained similar across depths. However, transplanted neurons were more uniformly distributed across depths during late sessions, with increased proportions of task-modulation neurons across all depths compared to early. We have now included this analysis in results and Figure S1c.

7) It is not clear whether the newly formed blood vessels are “functional”, as claimed by the authors. For instance, they could be not fully mature and leaking; I suggest the authors could evaluate the permeability of the vessels to confirm their functionality. Also, it would be interesting to evaluate if blood vessel density correlates with behavioral recovery.

Thank you for clarifying if the blood vessels are functional. In our experience with stroke and blood flow imaging, we have found that blood vessels in peri-lesional cortex are leaky for up to 2 weeks after a stroke (due to BBB disruption). If an animal is given an i.p. injection of the fluorescein dye during this period, the dye leaks into the PLC within 45 mins of i.p. injection. This can be seen as hyperintense signal in most of the imaging window, even in areas without a visible blood vessel. Taking this into consideration, we performed the blood flow imaging in transplanted mice 60 minutes after the i.p. injection to allow enough time for dye to leak out. We did not observe any leakage in the transplanted mice at 14 weeks post-implantation. Additionally, leakage of the fluorescein dye will be evident in post-mortem brain tissue, but we did not observe any such signals. This suggests that the blood vessels in the graft are functional, with intact BBB.

This is of course a very interesting question. Evaluating the relationship between blood vessel density and behavioral recovery is difficult in these stroke mice since blood flow imaging was not chronically performed due to technical challenges as described above. However, it may be possible in the future to conduct monitoring during the middle and later stages of recovery.

8) Since the approach presented here allow for a longitudinal evaluation of blood vessels, it would be interesting for the reader to have a timeline of their formation: are they already formed two weeks after stroke? Do they form during the training weeks? Does it correlate with behavioral recovery?

We agree with the reviewer that longitudinal evaluation of the blood vessels will be an interesting approach for evaluating the rate of angiogenesis and how it effects behavioral recovery. However, as mentioned above in 7), leakage of the dye from blood vessels at early stages is an important caveat that can hinder successful imaging of transplanted neurons by contaminating signals and increasing inflammation in the injured tissue. To avoid these potential conflicts for this study, we performed blood flow imaging at the late stages. Additionally, our previous histological work shows that when an ECM hydrogel is implanted in a stroke cavity, endothelial cells are dispersed within the ECM without proper vessel structures at 2 weeks (Ghuman et al 2018). At 3 months, blood vessels could be seen in the implanted grafts that were identical to host vessels. We hope that our platform can eventually optimize the period of monitoring that is possible and then ask this fundamental question.

Minor

points:

1) I could not find the total number of neurons that are (or are not) task modulated, only relative values (%). Please add this information.

We have added a plot with total number of task modulated neurons in Figure S1b. Number of task-modulated neurons monitored within the imaging window increased three-fold over the course of the training.

2) The legend at the end of Figure S1 states that the host tissue surrounding the transplant consists of the pre-motor cortex. This is true for just one side. Please correct.

Thank you for noticing this detail. We have now corrected this.

3) Figure 3, panel C: for clarity, please use a different color for healthy and early training groups.

We have now updated the color for healthy training group.

4) Lines 195-6: “This allowed us...based on their functional interaction with the perilesional cortex”. This claim is a guess, but it is not supported by the data.

We have now edited the manuscript to reflect this change that these interactions are not necessarily functional.

5) Please add average and error (SD o SEM) to quantify the distance of the cells to the blood vessels.

We have now added additional statistical information. To quantify distance of cells to blood vessels, we used the median, and 25th and 75th percentiles (q1 and q3) as a boxplot with whiskers showing data points greater than $q3 + 1.5 \times (q3 - q1)$ or less than $q1 - 1.5 \times (q3 - q1)$ to display statistical information. We have also added more details in methods.

6) Video 1 could be improved by adding a reference to the time point at which the reaching started. Also, the cells with task-related activity should be evidenced by a ROI and the relative traces of activation could be shown on the side of the video.

Video 1 is a 20 sec time-lapsed recording, accelerated to show the activation patterns of transplanted neurons over a 40 sec period. Since the reach-to-grasp movement is extremely rapid (< 250 ms), it is not clearly captured in this time lapse video. However, we have now edited Video 2 to include task-related ROIs and their traces.

7) Video 4: please show the cells as they appear along the depth together with the blood vessels.

We have now edited Video 4 to include the depth information.

8) Please comment on the fact that the expression of GCaMP could change over time, with more positive cells over the weeks. Do you see an increased fluorescence over the weeks?

We appreciate the thoughtful comment. To test if the fluorescence levels changed over recording periods, we measured the average peak amplitude within a recorded session and normalized it by dividing with noise level (i.e. residual activity) for that session. We observed no significant fluorescence change over time during three month period (One-way ANOVA: $p=0.11$, $R^2=0.47$, paired t-test: non-significant results for all pairs). We have now added these details in Methods and Figure S3b.

9) In the discussion, please comment on the importance of glial and endothelial cells in the transplant. Would the results change if only neurons were transplanted?

We recognize the importance of glial and endothelial cells in our transplant and have now amended our discussion to reflect their important role in supporting graft survival and function.

10) Does the turbidity/scattering of the transplant change over the weeks? Please add this information in the methods section.

We have now incorporated this information in the methods section. Besides changes in the cell maps during early phase (as described in Reviewer #4, Comment 3), we did not observe any changes in the turbidity/scattering of the transplant over the weeks. This is also evident from the average fluorescence plot in Fig. S3b showing no significant changes in fluorescence levels from active cells over the weeks.

Reviewer #4 (Remarks to the Author): Ghuman et al presented a toolbox for long-term tracking and monitoring of transplanted embryonic neurons in adult mice after cortical injury. Using this platform, they demonstrated that transplanted neurons were capable to integrate into the damaged cortical area and formed healthy networks. The results look very promising but I have several questions listed below:

1. I have some general questions about the destiny of transplanted embryonic cortical cells. Were GCaMP6f+ transplanted neurons developed into both excitatory neurons and inhibitory neurons? What was the survival rate after transplantation? Were the total survival neurons reduced with time? Can transplanted neurons migrate into other brain areas?

Thank you for requesting more information regarding the destiny of transplanted neurons. We have now incorporated Figure 7 into our manuscript to include this histological data. We show that the transplanted cells being monitored in live animals express markers for both inhibitory neurons (GFP+, GAD65/67+) and excitatory neurons (GFP+, GAD65/67-), and extend their axons into host peri-lesional and sub-cortical areas. Also, we noticed that majority of these transplanted neurons stayed within the stroke cavity after 3 months, with sparse migration into the peri-lesional cortex suggesting poor migration capacity of transplanted neurons in an adult brain.

Although we were not able to histologically confirm the survival rate in post-mortem brain tissue due to technical challenges (i.e. in most cases, removal of the implanted lens secured onto the skull by dental cement disrupted the tissue within and around stroke cavity), we did not observe any significant drop in total number of monitored neurons in live animals for 3 months. This can be seen in total number and proportions of modulated neurons in Fig. 2E and Fig. S1.

2. The authors indicated that adult host mice were transplanted with embryonic cortical cells from either Syn1-GCaMP6f or CaMKII-GCaMP6f mice. It seems that the authors did not further distinguish these two population for all of their analysis. I am wondering why. Cortical cells from Syn1-GCaMP6f mice might have more inhibitory neurons being labeled with GCaMP6f than that of CaMKII-GCaMP6f mice. This might affect the calculation for % of task modulation neurons.

We appreciate this comment. To evaluate any differences in observed task-related neural dynamics in transplanted mice with Syn and CaMKIIa promoter, we measured task modulated populations across both groups and found no significant differences (unpaired t-test: (early) $p = 0.73$, (mid) $p = 0.41$, (late) $p = 0.67$). We have amended our results to reflect this finding.

3. Were panels (E, M, L) in Figure 2C from the same mouse? If so, I am wondering why the cell maps changed so dramatically. If not (i.e., the panels were from three different mice), the authors should show the representative images from the same mouse to demonstrate increase of task-modulated neurons with time.

Thank you asking about this. Indeed, the cells maps in Fig. 2C are from the same mouse. Although this specific mouse had a more dramatic change in cell maps from early to mid sessions compared to the other mice, similar signs of tissue remodeling during early phase (< 5 WPI) could be seen in other mice. We believe these changes are a result of tissue remodeling within the stroke cavity, as well as peri-lesional tissue that follows a stroke. In this study, these changes were evident with a gradual improvement in the clarity of the imaging plane (patches of dark fog/possibly blood eventually cleared by immune cells) during

1-3 WPI, and emergence of active neurons during 2-4 WPI. By 5 WPI, active remodeling is mostly finished, with minor changes in cell maps during middle and late phases. This active remodeling during the early phase led to a significant drop in the total numbers of neurons that could be tracked between the early-late phases as shown in Fig. S3.

4. Fig 4e should include a health control (H) for comparison (as in Figure 2e and Figure 3b, 3c).

Thank you for inquiring about the changes in task-related correlations in tracked neurons from healthy mice. We have now included results for both Fig. 4e and Fig. 4f in Fig. S3 to show that thy1 neurons show similar trends of increasing task-related covariations, with broadening of correlation distributions similar to transplanted mice.

5. Figure 2e, figure legend indicated “n = 6 transplanted mice, n= 3 neurotypical mice”. Are there only a total of 6 transplanted mice being measured at three different time points (E, M, L)? The n number seems too low.

We appreciate this comment. We aim for our manuscript to be a proof-of-principle for long-term monitoring. The n=6 is fairly typical for work done in the field of neural dynamics. We note that in all 6 animals we were able to successfully monitor the cells over a 3 month period. The monitoring was actually done in a more continuous manner (~2-3x per week in early and mid sessions, and 1x per week in late). Thus, the totally number of sessions is ~20. We think the power of this approach is the repeated sampling in the same animal; this will allow much greater transparency into changes in the cavity and eventual understanding of causal contributions to recovery.

6. What is R square for Figure 2f?

Thank you for pointing out this missing information, we have now added R^2 value for Figure 2f.

REVIEWERS' COMMENTS

Reviewer #1 (Remarks to the Author):

I am satisfied that the authors took comments regarding the identity of the grafted cells into consideration. However, I am not satisfied that they present just normal fluorescent images without proper confocal validation of the morphology and identity of the cells. This needs to be done.

Reviewer #2 (Remarks to the Author):

These revisions address the previous concerns of this reviewer. They make for a stronger manuscript, and one in which now the direct effects of activity can be factored into an understanding of the transplanted cell network activity.

Reviewer #3 (Remarks to the Author):

The authors replied to my comments.

Reviewer #4 (Remarks to the Author):

All my concerns have been addressed by the authors.

REVIEWERS' COMMENTS

Reviewer #1 (Remarks to the Author):

I am satisfied that the authors took comments regarding the identity of the grafted cells into consideration. However, I am not satisfied that they present just normal fluorescent images without proper confocal validation of the morphology and identity of the cells. This needs to be done.

Author's response: Thank you for your comment. We have now moved the normal fluorescence microscopy images and figure into supplementary, and replaced the main figure 7 with confocal images.

Reviewer #2 (Remarks to the Author):

These revisions address the previous concerns of this reviewer. They make for a stronger manuscript, and one in which now the direct effects of activity can be factored into an understanding of the transplanted cell network activity.

Reviewer #3 (Remarks to the Author):

The authors replied to my comments.

Reviewer #4 (Remarks to the Author):

All my concerns have been addressed by the authors.